# Composites of Nucleic Acids and Boron Clusters (C_2_B_10_H_12_) as Functional Nanoparticles for Downregulation of EGFR Oncogene in Cancer Cells

**DOI:** 10.3390/ijms22094863

**Published:** 2021-05-04

**Authors:** Damian Kaniowski, Katarzyna Ebenryter-Olbińska, Katarzyna Kulik, Justyna Suwara, Wojciech Cypryk, Agata Jakóbik-Kolon, Zbigniew Leśnikowski, Barbara Nawrot

**Affiliations:** 1Centre of Molecular and Macromolecular Studies, Polish Academy of Sciences, Sienkiewicza 112, 90-363 Lodz, Poland; dkanio@cbmm.lodz.pl (D.K.); kebenryt@cbmm.lodz.pl (K.E.-O.); kpieta@cbmm.lodz.pl (K.K.); jmilczar@cbmm.lodz.pl (J.S.); wcypryk@cbmm.lodz.pl (W.C.); 2Department of Inorganic, Analytical Chemistry and Electrochemistry, Faculty of Chemistry, Silesian University of Technology, Krzywoustego 6, 44-100 Gliwice, Poland; agata.jakobik-kolon@polsl.pl; 3Laboratory of Medicinal Chemistry, Institute of Medical Biology, Polish Academy of Sciences, Lodowa 106, 92-232 Lodz, Poland; zlesnikowski@cbm.pan.pl

**Keywords:** antisense oligonucleotide, boron cluster, nanostructure, EGFR, macrophages, cellular uptake

## Abstract

Epidermal growth factor receptor (EGFR) is one of the most promising molecular targets for anticancer therapy. We used boron clusters as a platform for generation of new materials. For this, functional DNA constructs conjugated with boron clusters (B-ASOs) were developed. These B-ASOs, built from 1,2-dicarba-*closo*-dodecaborane linked with two anti-EGFR antisense oligonucleotides (ASOs), form with their complementary congeners torus-like nanostructures, as previously shown by atomic force microscope (AFM) and transmission electron cryo-microscopy (cryo-TEM) imaging. In the present work, deepened studies were carried out on B-ASO’s properties. In solution, B-ASOs formed four dominant complexes as confirmed by non-denaturing polyacrylamide gel electrophoresis (PAGE). These complexes exhibited increased stability in cell lysate comparing to the non-modified ASO. Fluorescently labeled B-ASOs localized mostly in the cytoplasm and decreased EGFR expression by activating RNase H. Moreover, the B-ASO complexes altered the cancer cell phenotype, decreased cell migration rate, and arrested the cells in the S phase of cell cycle. The 1,2-dicarba-*closo*-dodecaborane-containing nanostructures did not activate NLRP3 inflammasome in human macrophages. In addition, as shown by inductively coupled plasma mass spectrometry (ICP MS), these nanostructures effectively penetrated the human squamous carcinoma cells (A431), showing their potential applicability as anticancer agents.

## 1. Introduction

Epidermal growth factor receptor (EGFR) is one of the most promising molecular targets for anticancer therapy [1]. This receptor is a transmembrane protein located either on the cell surface or in a diffused form in the cytoplasm [2]. EGFR consists of three major functional domains: an extracellular ligand-binding domain, a hydrophobic transmembrane domain, and an intracellular enzymatic domain [3,4,5,6]. Activation of EGFR by its ligands initiates several signal transduction cascades, mainly the c-Jun N-terminal kinase (JNK), protein kinase B (Akt/PKB), and mitogen-activated protein kinase (MAPK) pathways, leading to DNA synthesis and cell proliferation [7,8,9,10]. The proteins involved in these pathways modulate invasiveness, such as cell migration, adhesion, and proliferation. EGFR is expressed in healthy cells at a low level and plays an essential role in normal cell signaling [11]; however, in cancer cells, mutations in EGFR lead to its overexpression. A high density of EGF receptors leads to uncontrolled cell proliferation, angiogenesis, invasion, metastasis, and inhibition of apoptosis [12,13,14,15,16]. Moreover, EGFR is a negative cancer prognostic factor [17].

The identification of EGFR as an oncogene has led to the development of anticancer therapeutics directed towards it [18]. The use of monoclonal antibodies targeting EGFR is the most common but not a very effective pharmacological approach, as EGFR is present not only in the external membranes of cells but also in intracellular membranous organelles, e.g., mitochondria, endoplasmic reticulum, endosomes, or the cell nucleus, rendering it inaccessible to antibodies, resulting in therapy failure [19,20]. Another anti-EGFR therapy is based on the use of low molecular weight compounds that bind to the tyrosine kinase domain of the epidermal growth factor receptor and block its activity [21,22].

The FDA (Food and Drug Administration) and EMA (European Medicines Agency) already approved several low molecular tyrosine kinase inhibitors (gefitinib, erlotinib, afatinib, and osimertinib) as first line treatment for sensitive EGFR-mutant patients with non-small-cell lung cancer (NSCLC) [23]. Although third-generation EGFR inhibitors exhibit limited side effects [24], there is still a need to develop innovative therapeutic molecules capable of overcoming the challenges faced by existing EGFR inhibitors.

Recent years have seen the rapid development of drugs for gene therapy based on nucleic acid scaffolds and the exploitation of the intracellular enzymatic machinery to deplete messenger RNA identified as a therapeutic target. During the past 2 decades, variously modified antisense oligonucleotides (ASOs), short interfering RNAs (siRNAs), and aptamers, often decorated with delivery molecules, have been successfully developed and characterized as inhibitors of EGFR gene expression at the preclinical level. Among them, sugar backbone-modified antisense oligonucleotides were able to inhibit proliferation, induce apoptosis, and cooperate with cytotoxic drugs in human cancer cell lines [25,26,27,28] and in xenograft mice [29]. Phosphorothioate-modified ASOs (PS-ASOs) have been shown to downregulate EGFR via RNase H activation [30]. Short interfering RNAs downregulating the expression of epidermal growth factor receptors have been shown to enhance the chemosensitivity of well-established anticancer compounds (cisplatin, 5-fluorouracil, and docetaxel) [31] and radiosensitivity [32] in various cancer cells, including head and neck squamous cell carcinoma (HNSCCs). The anti-EGFR aptamer [33] induces selective apoptotic cell death and, upon structural optimization [the use of 2′-*O*-methyl (2′-OMe) units], its human serum stability and toxic potential are increased in breast cancer, liver cancer, and malignant glioblastoma cells [34].

In our previous research, we designed novel self-assembling DNA nanostructures bearing antisense oligonucleotides directed towards EGFR mRNA [35]. To date, various types of self-assembling DNA-based nanostructures have been developed for biomedical applications, including drug delivery, immunotherapy, diagnostics and molecular biology [36]. In vitro assays assessing the stability of small nanoconstructs consisting of a few oligonucleotides [37] and scaffold DNA origami structures [38] have demonstrated that these structures show drastic differences compared with their linear analogs. However, only a limited number of studies have shown the applicability of self-assembling nanostructures for the regulation of target disease-related gene expression, e.g., DNA-backboned bottlebrush nanostructures with high-density PEG (polyethylene glycol) side chains containing antisense overhangs directed towards KRAS (a Kirsten ras oncogene homolog from the mammalian ras gene family) mutants in lung cancer cells [39] and self-assembling RNA nanoparticles containing functional siRNAs directed towards model EGFP (enhanced green fluorescent protein) mRNA [40].

In our approach, we utilized tripeds of 1,2-dicarba-*closo*-dodecaborane (1,2-DCDDB, C_2_B_10_H_12_) conjugated with antisense oligonucleotides, which, by complexation with their complementary counterparts, can self-assemble into higher-order structures and are active EGFR gene silencers in a model system [35]. These complexes after enrichment with ^10^B atoms have the potential to exert a dual anticancer effect: antisense EGFR downregulation and delivery of boron atoms for boron neutron capture therapy (BNCT) [41]. In this work, we analyzed the assembly of tripeds **1** and **2** in detail and screened the nucleolytic stability and silencing properties of nanostructures **1**/**2** towards endogenous EGFR in the human squamous carcinoma cells (A431). We also investigated whether nanostructures have pro-inflammatory properties by activating the inflammasome-mediated IL-1β production in human primary monocyte-derived macrophages.

## 2. Results

### 2.1. Chemistry

Two bis-functionalized tripeds, **1** and **2**, that are conjugates of 1,2-dicarba-*closo*-dodecaborane (1,2-DCDDB, C_2_B_10_H_12_) and two short complementary DNA strands (Figure 1) were used. Tripeds **1** and **2** were synthesized as described previously [35] by elongation of LCA CPG (long chain alkylamine controlled pore glass beads)-immobilized-1-(2-*O*-succinylethyl)-9,12-bis (3-*O*-tritylprop-1-yl)-1,2-dicarba-*closo*-dodecaborane with 22-meric oligonucleotide chains by routine solid phase synthesis and a subsequent deprotection work up. The 5′-d(TTT CTT TTC CTC CAG AGC CCGA)-3′ sequence of the oligonucleotides attached to the boron cluster in **1** is complementary (antisense) to the coding fragment of the EGFR mRNA, while the 5′-d(TCG GGC TCT GGA GGA AAA GAAA)-3′ sequence of the oligonucleotide in triped **2** is complementary by the Watson–Crick model to the oligonucleotide strands of **1**. For these studies, 5′-fluorescein tagged **1** and **2** (**FL-1** and **FL-2**) were obtained. Spectral and chromatographic characteristics are given in Appendix A and Appendix A.

### 2.2. Assembly of Complexes of **1** and **2**

Tripeds **1** and **2** were designed to form 22-nucleotide (nt) fragments that assembled into antiparallel 22-base pair (bp) DNA duplexes. We have previously shown that a 1:1 molecular ratio mixture of tripeds **1** and **2** can be successfully assembled into ring-shaped nanostructures of various sizes containing typically 6÷12 tripeds, as demonstrated by AFM and cryo-TEM imaging [35]. The distribution of the torus-like nanostructures in the obtained images was assessed by the IC-Measure program (https://www.theimagingsource.com/ (accessed date 15 April 2021) version 2.0.0.133, 2018). Here, we describe the electrophoretic conditions under which particular ^32^P-radioactively labeled complexes were visible as separate spots allowing the determination of each product content in solution. As shown in Figure 2A, all the complexes that formed had lower mobility than the reference triped **1**. Autoradiographic analysis shows four dominant products accounting for approximately 9, 21, 51, and 19%, corresponding to nanostructures built from 44-, 88-, 132-, and 176-base pair DNA components, which corresponds to complexes formed of total 2, 4, 6, and 8 tripeds **1** and **2**, respectively (Figure 2A). This result is consistent with our earlier assessment based on AFM and cryo-TEM analyses [35], which showed the largest population of nano-objects in the range of 21–40 nm in diameter, corresponding to nanostructures consisting of 6 to 12 tripeds **1** and **2**.

We also confirmed that the annealing efficiency of **FL-1** and **FL-2** (1:1 molar ratio) was similar (data not shown) to that of the parent models (see Figure 2A and Appendix A).

### 2.3. Nucleolytic Stability of Complexes of **1**/**2**

The nucleolytic stability of complexes **1**/**2** (Figure 2B) and ASO-22, which is an oligonucleotide fragment of triped **1** (Figure 2C), was investigated in DMEM (Dulbecco’s Modified Eagle Medium) cell culture medium (left panels) and in human squamous cell carcinoma cell lysate (A431, right panels) in the time range of 0, 12, 24, and 48 h at 37 °C. The degradation mixtures were analyzed by nondenaturing polyacrylamide gel electrophoresis (PAGE) and visualized either by Stains-All staining (Figure 2B,C left panels) or by autoradiography (Figure 2B,C right panels). Both the ASO-22 oligonucleotide and complexes **1**/**2** were shown to be stable in the cell culture medium during the follow-up period of 48 h, while the samples were slowly hydrolyzed in the A431 cell lysate. Only approximately 35% of the intact ASO-22 was present in the reaction mixture after 48 h of incubation. In contrast, approximately 48% of the **1**/**2** complexes were still present in the reaction mixture, indicating their moderately increased stability under the tested conditions. As shown in the plot in Figure 2D, the half-life (t_1/2_) of double-stranded DNA nanostructures **1**/**2** (t_1/2_ = 45 h) was twice as long as that of single-stranded DNA oligonucleotides (t_1/2_ = 21 h).

### 2.4. EGFR-Targeted Gene Silencing Activity of Nanostructures **1**/**2**

#### 2.4.1. In Vitro RNase H-Assisted RNA Cleavage

Antisense oligonucleotides exert their gene silencing activity either by sequence-specific hybridization to target mRNA molecules and recruitment of RNase H, which cleaves the target RNA, as well as by other mechanisms involving the steric hindrance of mRNA ribosomal activity, or by altering mRNA maturation [42]. Here, we tested the ability of boron-cluster-embedded ASO (**1**) in the duplex with a complementary 5′-[^32^P]-labeled RNA fragment to trigger RNase H activity in comparison to the nonmodified reference ASO-22 (Figure 3A right and left panels, respectively). We used *E coli* RNase H, which is more available than the human enzyme, and its sequence preferences are nearly identical to those of its human counterpart [43]. The first hydrolysis products were observed for both screened reactions after 15 min, and the entire RNA substrate was degraded entirely in each case after 30 min (Figure 3A). Interestingly, in the case of ASO-22, two main 5′-products of 5′-[^32^P]-RNA cleavage were observed (9 and 7 nt). Only traces of shorter 6 nt long product was visible. These results indicate that RNA was cleaved within the internucleotide linkages marked by the arrows in the 5′-pUCG GGC U↓CU ↓GGA GGA AAA GAAA-3′ sequence, that is, between U and C or U and G units, respectively. The prolonged incubation time of the cleavage reaction up to 240 min (Appendix A) resulted in an increase in the content of the 7 nt product, which suggests that RNase H is further activated by the duplex of 9 nt RNA with ASO-22, resulting in the shorter (7 nt) radioactive product. In the case of triped **1**, which contains two ASO-22 strands, the substrate RNA was completely cleaved in the first 15 min, and one 9 nt product was then released. After the next 15 min, three 5′-[^32^P]-RNA cleavage products (Figure 3A, right panel) of 9-, 7-, and 6-nucleotides were present, with the shorter product being prevalent. After 60 min, only the shortest 6 nt radioactive product was observed, and no other product appeared after the longer incubation time (up to 240 min) (Appendix A). Thus, we also observed further RNase H activation by the duplexes of preliminary 9 nt and 7 nt RNA products with **1**; however, the preferred cleavage occurred after the sixth nucleotide (5′-pUCG GGC ↓UCU GGA GGA AAA GAAA-3′), which is between a C and U. The obtained results also suggest that the rate of hydrolysis of **1**/**2** is slightly faster than that of ASO. This experiment shows the sufficient efficiency of the tested nanostructures for duplex formation with the target RNA, resulting in RNase H activation and successful degradation of the target RNA.

#### 2.4.2. Downregulation of Endogenous EGFR mRNA in A431 Cells Monitored by Western Blotting

To test the silencing activity of nanostructures **1**/**2** towards the endogenous mRNA of EGFR protein, we used human squamous cell carcinoma cells (A431) expressing higher levels of EGF receptor compared to HeLa (human cervical carcinoma) and MCF-7 (human breast cancer) cell lines [35]. Thus, A431 cells were transfected with either ASO-22 or the **1**/**2** complex (obtained by annealing of the mixture of **1** and **2** in a 1:1 molar ratio) in the presence of Lipofectamine 2000 at concentrations of 0, 5, 25, 50, and 100 nM. The level of cellular EGFR protein after a 48 h incubation was assessed by sodium dodecyl sulphate (SDS) PAGE electrophoresis of protein lysates and EGFR-specific immunoblotting (Figure 3B,D, respectively). These results, which were quantified and plotted in Figure 3C,E, respectively, demonstrate that the level of EGFR protein drops gradually down to 30% (70% silencing, *p* ≤ 0.0001) for **1**/**2** used at a 100 nM concentration, while ASO-22 in the same conditions is able to decrease the EGFR level by less than 50% (*p* ≤ 0.05). These results confirm the high gene expression inhibitory activity of B-ASO nanostructures **1**/**2** directed towards endogenous mRNA of EGFR in the A431 tumor cell line, higher than that of the non-modified ASO-22.

#### 2.4.3. Exogenous EGFR mRNA in A431 Cells Monitored by Fluorescence Imaging

Previously, we confirmed the efficient silencing activity of **1**/**2** towards exogenous EGFR mRNA in an EGFR-EGFP/RFP (EGFR-enhanced green fluorescent protein/red fluorescent protein) dual fluorescence assay (DFA) performed in MCF-7, A431, and HeLa cells and the fluorescence measurements performed on the plate reader [35]. In the present work, the silencing activity of **1**/**2** and ASO-22 was confirmed in a DFA in A431 cells by imaging with fluorescence microscopy (see Appendix A). In this experiment, the pair of plasmids coding gene of the EGFR-EGFP fusion protein and of the control RFP protein and the tested compounds at concentrations of 50 and 100 nM were used for the Lipofectamine 2000-assisted transfection of A431 cells, and expression of the EGFR-EGFP and RFP fluorescent proteins after 48 h incubation was monitored by microscopic analysis of live cells. The cells expressing both plasmids transfected with 100 nM nonsense ASO-C (Appendix A) were used as the control [44,45]. Interestingly, the silencing effect of **1**/**2** was as efficient as ASO-22 used at the same concentration, suggesting that nanostructures **1**/**2** inhibit the expression of EGFR gene in the exogenous EGFR-EGFP/RFP dual fluorescence model (Appendix A) as well as in the endogenous A431 cellular system (Figure 3B–E), and this process is hindered neither by the presence of boron clusters nor by the originally double stranded structure of higher-order complexes.

### 2.5. Microscopic Analysis of the Localization of FL-1/FL-2 Nanostructures in Cancer Cells

As previously demonstrated, fluorescein-labeled *closo*-carboranyl oligomeric phosphate diesters are relatively stable under either moderately acidic or basic conditions, and the fluorescein group is unlikely to be released from oligomers under physiological conditions [46]. Knowing this, we prepared fluorescent derivatives of both tripeds **1** and **2** by labeling their 5′-ends with the 6-FAM fluorescent tag and used them to assess the cellular localization of nanostructures **1**/**2**. The procedure for the synthesis of **FL-1** and **FL-2** is described in Section 4.1 and the chromatographic and structural analyses of **FL-1** and **FL-2** are provided in the Supplementary Information (Appendix A). The annealing efficiencies of **FL-1** and **FL-2** (complexed in a 1:1 molar ratio) and the distribution of the dominant population of **FL-1/FL-2** nanostructures were similar to those of the complexes of parent tripeds **1** and **2**, as confirmed by electrophoretic analysis in a nondenaturing gel (data not shown). In this experiment, the localization of the **FL-1/FL-2** nanostructures was monitored in A431 cancer cells expressing high levels of EGF receptors, and in HeLa and MCF-7 cancer cells with a lower EGFR density. The cells were transfected with complexes of **FL-1/FL-2** (100 nM) in the presence of Lipofectamine 2000, and after 6 h incubation, the cells were visualized by green fluorescence imaging (Figure 4, row 2). In parallel, the nuclei of the tested cells were stained with DAPI, and the endoplasmic reticulum membranes were stained with endoplasmic reticulum (ER)-Tracker Red (Figure 4, row 3). Careful analysis of the localization of green B-ASO nanostructures with blue nuclei (row 4) or with red ER (row 5) indicates that fluorescent B-ASOs are located in the cell cytoplasm, where the ASOs exert their silencing activity. This process probably involves, at least in part, helicase-assisted unwinding of nanostructures **1**/**2** to their individual strands (and no spontaneous dissociation as T_m_ of **1**/**2** is approximately 70 °C [35]), followed by the association of oligonucleotides of triped **1** with the target mRNAs and triggering RNase H cleavage. Moreover, the green signals overlapping the DAPI signal in the fourth row and the green signals next to the ER signals in the fifth row suggest that **FL-1/FL-2** may also be present in the nuclei. There are studies on the PS-ASOs that showed the nuclear localization of antisense oligonucleotides [47]. As shown in the DAPI and FITC filter test (see Appendix A), the observed green signals in the nuclei are not from excitation of the fluorescein chromophore by the blue wavelength emitted by DAPI.

### 2.6. Influence of Nanostructures 1/2 on the Cancer Cell Phenotype and Migration Rate

EGFR plays a key role in cell growth, intercellular communication, and metastasis of neoplastic cells. Inhibition of the expression of the *EGFR* gene can effectively inhibit the growth and intercellular communication [48]. To determine whether ASO-22 and nanostructures **1**/**2** can exert similar effects, these compounds were transfected into HeLa cells at concentrations of 0, 50, and 100 nM in the presence of Lipofectamine 2000, and after 48 h of incubation, the phenotypic changes in live cells were microscopically analyzed. As shown in Figure 5A, characteristic phenotypic changes of HeLa cells including changed cellular shape and the loss of intercellular communication junctions were observed in cells transfected with either **1**/**2** and with ASO-22 as compared to control cells transfected with nonsense ASO-C (100 nM) (*p* ≤ 0.0001) [44,45]. This result indicates an intracellular activity of anti-EGFR antisense oligonucleotides present in the **1**/**2** nanostructures, similar to free ASO-22.

A wound healing assay was used to study the changes in the cell migration rate upon incubation of the cell culture with ASO-22 or with nanostructures **1**/**2** at concentrations of 0, 50, and 100 nM. The images of the scratch on the HeLa cell monolayer were microscopically analyzed at 0 h and 48 h of incubation (Figure 5B images), and the number of cells in the wound region was counted (Figure 5C). These results show strong inhibition of HeLa cell migration after administering unmodified ASO-22 and **1**/**2** nanostructures, although the effect was much more pronounced for the latter EGFR-directed antisense constructs. Thus, nanostructures **1**/**2** at a concentration of 50 nM showed a twice slower wound repair rate (decreased the cell number by approximately 60%) than the oligonucleotide reference ASO-22 (50 nM, transfected with ASO-C (100 nM)) (decreased the cell number by approximately 30%) compared to the control cells. This effect was much larger for nanostructures **1**/**2** used at 100 nM (cell migration rate inhibition of approximately 80%).

### 2.7. Cell Cycle Analysis upon 1/2 or ASO-22 Transfection

To analyze the influence of nanostructures **1**/**2** and the reference oligomer ASO-22 on the cell cycle, A431 cells were transfected with tested oligomers in the presence of Lipofectamine 2000, and fluorescence-activated cell sorting (FACS) analysis was performed after 48 h of incubation (Figure 6). A431 cells cultured with Lipofectamine 2000 were used as a control. The obtained data indicate that the unmodified ASO-22 antisense oligonucleotide at a concentration of 100 nM increased the ratio of transfected cells in the G1 (79%) and S (19%) phases relative to the control A431 cells in the G1 (69%) and S (16%) phases, while nanostructures **1**/**2** (100 nM) slightly changed the proportion of A431 cells in the G1 phase (73%) but increased the proportion of A431 cells in the S phase (25%). Under the same conditions, both ASO-22 and **1**/**2** strongly inhibited the ratio of cells in the G2 phase to approximately 2% compared to the control (15%). The obtained result suggests that the cell cycle of A431 cells upon transfection with ASO-22 or nanostructures **1**/**2** is arrested in the S phase and prevents the transition of cells into the G2 phase and of cell division.

We also checked the influence of free boron clusters on the cell cycle by treating A431 cells with 1,2-dicarba-*closo*-dodecaborane (1,2-DCDDB), which is a component of nanostructures **1**/**2**, and a control boron cluster, a metallacarborane [(3,3′-iron-1,2,1′,2′-dicarbollide) (-1)]-ate [Fe(C_2_B_9_H_11_)_2_]^−^ (FESAN)) (50 and 100 nM). The 1,2-DCDDB boron cluster (100 nM) increased the content of cells in the S phase to 24% and strongly decreased the content of cells in the G2 phase to 2% compared to the control cells (16% in S phase, 15% in G2 phase). The changes observed in a cell cycle phases ratio induced by ASO-22 and **1**/**2** follow a very similar pattern to 1,2-DCDDB, with small changes only in the S (increased by 3–14%) and G1 phases (up to 10%) compared to the control cells, while the abundance of the G2 phase remains very low in all experiments (2–3%) compared to 15% for the control cells. The most significant changes in the distribution of the cell cycle phases were observed for FESAN-treated A431 cells (100 nM). Here, the ratio of cells in the S phase increased to 74% and that in the G1 and G2 phases decreased to 24% and 2%, respectively.

Colchicine and quercetin were used as positive controls for A431 cells. Earlier data reported that colchicine induces cell cycle arrest at the G2/M phase in MCF-7 cells [49], while quercetin causes cell cycle arrest in the S-phase [50]. Our data confirmed these reports, as colchicine significantly inhibited cell viability at a concentration of 50 nM by inducing cell cycle arrest at the G2/M phase (74%) in A431 cells. Quercetin at the same concentration resulted in cell cycle arrest by increasing the proportion of A431 cells in the S-phase to 34%. FACS analysis with confidence interval (CI) and the mean with standard deviation (±SD) statistics is given in Appendix A.

### 2.8. Nanostructures 1/2 Do Not Activate the Inflammasome in Human Macrophages

Local inflammation is frequently observed during tumorigenesis. In most cancers, inflammation enhances tumor development and malignant progression [51] as well as metastasis [52]; therefore, potential anticancer agents should not trigger inflammatory response. To learn whether nanostructures **1**/**2** exert inflammatory responses, we stimulated LPS-primed human primary monocyte-derived macrophages with nanostructures **1**/**2**, oligonucleotide ASO-22 and boron clusters 1,2-DCDDB and FESAN for 16 h and studied the release of major NLRP3 inflammasome-dependent pro-inflammatory cytokine IL-1β. As positive controls for IL-1β release, cells were stimulated with nigericin, an NLRP3 inflammasome-activating ionophore toxin derived from *Streptomyces hygroscopicus* [53]. Because the experiment was carried out on primary cells derived from two independent donors, the levels of IL-1β secretion are expressed as a % of nigericin-induced IL-1β. As shown in Figure 7A, neither of the studied nanostructures activated the inflammasome within 16 h. In contrast, FESAN treatment induced a significant increase in the level of IL-1β secretion (25–60% of nigericin control) (Figure 7A). To verify that cells were viable throughout stimulation, we performed a cell viability assay using 3-(4,5-dimethylthiazol-2-yl)-2,5-diphenyltetrazolium bromide conversion into formazan salt (MTT assay). As shown in Figure 7B, neither ASO-22 nor nanostructures **1**/**2** interfered with cells viability, while 1,2-DCDDB and FESAN reduced it by approximately 25%.

### 2.9. Nanostructures 1/2 Are Boron Atom Vehicles for A431 Cancer Cells

Finally, we investigated whether nanostructures **1**/**2** could effectively deliver boron atoms (B) to cancer cells for BNCT. Recent reports suggest that as many as 5–20 µg B/g cells in human melanoma cells are sufficient to exert a therapeutic effect in BNCT therapy [54]. In our experiments, we used A431 cells and nanostructures **1**/**2** at concentrations of up to 4 µM, which were delivered to the cells either by Lipofectamine 2000-assisted transfection or by free uptake. After 48 h of incubation at 37 °C, the cells were lysed, the boron atom content was determined with inductively coupled plasma mass spectrometry (ICP MS) [55,56], and calibration curves for both stable boron isotopes (^10^B and ^11^B) were used to assess the content of boron atoms delivered to the cells. As shown in Figure 8, nanostructures **1**/**2** were effective boron carriers when transfected into cells with the assistance of Lipofectamine 2000. The content of boron atoms in A431 cells, in this case, reached 9 µg B/g cells and 26 µg B/g cells for nanostructure **1**/**2** at 0.4 and 4 µM, respectively. For **1**/**2** delivered via free uptake, the content of boron atoms reached 6 and 13 µg B/g cells; however, these concentrations still appear sufficient for BNCT applications.

## 3. Discussion

In the approach described here, we designed molecules that offer advantageous, double therapeutic potential directed towards cancer cells. First, these molecules contain antisense DNA fragments and can exert inhibitory activity towards the target gene, and second, they contain boron clusters and thus can function as boron atom carriers for BNCT therapy. The 1,2-dicarba-*closo*-dodecaborane (1,2-DCDDB, C_2_B_10_H_12_) used here is a ball-shaped boron cluster with two flexible ortho-positioned hydroxypropyl functionalities situated at an angle of approximately 60°. These functionalities were loaded with two short oligonucleotide fragments (22-mers), antisense towards the mRNA of EGFR, forming the so-called triped **1**. The second triped **2**, of similar structure, contained a boron cluster with two attached oligonucleotides complementary to the ASO sequence in **1** (Figure 1). Annealing of these two tripeds at a 1:1 molar ratio allowed for precise spatial self-assembly, resulting in the formation of **1**/**2** higher-order complexes. In our previous work, using AFM and cryo-TEM microscopy [35], we defined the structure of the formed **1**/**2** complexes as torus-like nanostructures, with the dominant fraction being 21–40 nm in diameter, composed of 6÷12 tripeds (132-264-bp). Here, by electrophoretic analysis under nondenaturing conditions, we confirmed that in solution, four major **1**/**2** complexes are formed, with DNA strands containing 44-, 88-, 132-, and 176-bp in total (Figure 2A). Thus, the assembled nanostructures are cyclic dimers, tetramers, hexamers, and octamers containing 2, 4, 6, and 8 boron clusters linked by 22-bp DNA fragments, respectively (see Figure 2A). By autoradiographic analysis of the PAGE gels, the abundance of the particular complexes was determined as being approximately 9, 21, 51, and 19%, respectively. The ring-shaped hexamers, of approximately 20 nm in diameter, constitute the largest population among the formed complexes. Importantly, nanostructures of such a size may have the potential to spontaneously penetrate neoplastic tumors, as has previously been shown for nanoparticles with a diameter of approximately 30 nm, which can freely penetrate solid tumors with strong fibrosis and localize in intercellular spaces [57].

ICP MS studies, based on the measurement of the amount of boron in cancer cells, confirmed that nanostructures **1**/**2** applied to skin cancer cells (A431) at a concentration of 4 µM were able to penetrate the cell membrane without the transfection agent and provided 13 µg B/g cells, which corresponds to 2.6 µg ^10^B/g cells, as the natural abundance of this isotope is only approximately 20% (Figure 8). In contrast, when **1**/**2** nanostructures at the same concentration were transfected into A431 cells with the assistance of Lipofectamine 2000, 26 µg B/g cells (corresponding to 5.2 µg ^10^B/g cells) were found (Figure 8), although this transfection agent cannot be used in humans. Horiguchi et al. [58] have shown that a 5 ppm (5 µg ^10^B/g cells) concentration of ^10^B in cells is sufficient for effective BNCT therapy. Penetration of nanostructures through the phospholipid membrane without the transfection agent may also be partially supported by the neutral, hydrophobic 1,2-DCDDB moiety, which increases the lipophilicity of the entire molecule [35]. We believe that the obtained nanostructures with a diameter of approximately 20–25 nm (hexamers and octamers) or smaller may have the ability to penetrate tumor tissues and saturate them with a sufficient amount of boron necessary for BNCT therapy. Moreover, by lowering the level of endogenous EGFR in cancer cells, one can sensitize these cells to radiotherapy, which would lead to an enhancement of the therapeutic effect [59,60]. Importantly, the ASO sequences of triped **1** were designed to be effective silencers towards EGFR as well as towards EGFR variant III [61], playing a key role in cancer resistance to chemotherapy and radiotherapy.

In this work, we demonstrate that complexes **1**/**2**, due to their cyclic double-stranded structure, exhibited moderately increased nucleolytic stability in A431 cell lysates (Figure 2B) compared to the free antisense oligonucleotide ASO-22 (Figure 2C). The observed prolongation of the half-life of double-stranded DNA nanostructures **1**/**2** compared to single-stranded DNA oligonucleotides (t_½_ = 45 and 21 h, respectively, Figure 2D) may reflect the inferior accessibility of the nucleophile of DNA nuclease to a single-stranded oligonucleotide, which has unstacked bases compared to double-stranded helical DNA [62]. Moreover, due to the proven ability of 1,2-DCDDB to bind to albumin, the nucleolytic stability of the **1**/**2** nanostructures in vivo may be additionally enhanced and thus provide a longer biodistribution time in the blood stream and ensure enhanced nucleolytic protection [63].

Of note, the fluorescently labeled **1**/**2** (**FL-1/FL-2**) nanostructures after 6 h incubation are located in the cytoplasm in the region of the endoplasmic reticulum membranes of model tumor cells, including A431, HeLa, and MCF-7 cells (Figure 4), suggesting their accessibility to RNase H and the translational machinery in which EGFR mRNA is evidently engaged. This process involves, at least in part, helicase-assisted unwinding of nanostructures **1**/**2** to their individual strands, followed by the association of oligonucleotides of triped **1** to the target mRNAs and triggering RNase H cleavage activity. However, it cannot be ruled out that some amount of nanostructured fluorescent B-ASO is transported into the nuclei, as was already shown by some studies of antisense oligonucleotides [47]. The Viñas team has also shown that organotin dyes bearing boron clusters are localized mostly in the cytoplasm, but traces are also found in the cell nucleus [64].

Previously, we showed that nanostructures **1**/**2** reduced the exogenous EGFR level by more than 60% at a concentration of 200 nM after 48 h using DFA (EGFR-EGFP/RFP dual fluorescence assay) performed in MCF-7 (breast cancer), A431 (squamous carcinoma), and HeLa (cervical carcinoma) cells [35]. A similar effect was observed also for other B-ASO anti-EGFR inhibitors [45,65]. In this work, we assessed the ability of **1**/**2** to inhibit endogenous *EGFR* gene expression and reduce the level of EGFR receptors using Western blot analysis in A431 neoplastic cells. After 48 h, the **1**/**2** nanostructures at a concentration of 100 nM decreased the EGFR protein level by over 70% compared to the control (Figure 3D,E), suggesting that they are better silencers than ASO-22 (decrease in the EGFR by <50%, Figure 3B,C). We confirmed that the ASO strands in triped **1**, despite their attachment to the boron cage (1,2-DCDDB), are able to hybridize with the target RNA sequence and lead to RNase H-assisted RNA hydrolysis at a rate slightly higher than that of the unmodified ASO-22/RNA (Figure 3A). Possibly, the kinetics of the RNA cleavage upon binding to **1**/**2** is faster than that to ASO-22, which may result from favorable interactions of the boron cluster with the protein [62].

The other observed difference in this process was the cleavage site of the target 5′-[^32^P]-RNA. In the 5′-p*UCGGGC_6_U_7_C_8_U_9_GGAGGAAAAGAAA-3′ oligonucleotide, cleavage occurred after U_9_ and then after U_7_ in the presence of ASO-22, with only trace amounts of the product of cleavage after C_6_, while in the presence of **1**/**2**, cleavage occurred after the same units (see Figure 3A, cleavage after 30 min); however, in 60 min or longer (4 h) (Appendix A), only one product was observed as a result of the cleavage of p*RNA after C**_6_**. We cannot rationally explain this difference, although one can speculate that the boron cluster may exhibit affinity to both the protein and nucleic acid; therefore, the final cleavage product in both cases is p*UCGGGC_6_, which appears at a higher rate for the reaction with **1**/**2** compared to the reaction with ASO-22. Regardless of the result, this experiment shows the sufficient efficiency of the tested triped **1** for duplex formation with the target RNA, resulting in RNase H activation and successful degradation of the target RNA [66].

The silencing activity of nanostructures **1**/**2** towards endogenous EGFR was also demonstrated in the wound healing assay, in which approximately 80% inhibition of migration rate was observed in HeLa cells (100 nM, 48 h) (see Figure 5). The antiproliferative properties of the nanostructures were reflected by the A431 cell cycle arrest caused by an increased population of cells in the S phase and a decreased population in the G2 phase (Figure 6). In recent transcriptomic studies, it was observed that conjugates of small molecule compounds with a boron cluster greatly influence changes in the cell cycle of U87 glioblastoma cells, inhibiting the cell proliferation process and stimulating cells to enter the apoptotic pathway [67]. The aforementioned data suggest that the therapeutic effect of conjugates with a boron cluster can be achieved without the necessity of meeting the BNCT condition and irradiating tumor cells with a thermal neutron beam [67].

Boron is a regulator of the immune and inflammatory reactions and macrophage polarization, playing an important role in augmenting host defense against infection, with possible roles in cancer and other diseases [68]. Boron-containing compounds stimulate the production and secretion of nitric oxide, TNF-α, and IL-6, but also the secretion of IL-1β via inflammasome activation in macrophages, leading to acute inflammation in mice. Hyaboron, a boron-containing macrodiolide, has been shown to activate inflammasome-dependent IL-1β secretion by acting as potassium ionophore [69]. The mechanisms by which boron-containing compounds modulate immune responses, as well as discussion on the structure–activity relationship for each observed mechanism of action with respect to a production of cytokines, cell differentiation, proliferation, and antibody production were recently summarized by Romero-Aguilar K. et al. [70]. Therefore, we checked whether **1**/**2** are capable of activating the NLRP3 inflammasome. As shown in Figure 7A, nanostructures **1**/**2** containing 1,2-DCDDB did not stimulate primary macrophages isolated from two healthy human donors to produce IL-1β in contrast to FESAN, which induced IL-1β secretion at a level of approximately 30–50% of that of nigericin-induced IL-1β. These results show the lower pro-inflammatory properties of our compounds and demonstrate their improved potential safety in therapeutic application over boron carriers currently used in BNCT.

## 4. Materials and Methods

Chemicals were obtained from Aldrich Chemical Company (St. Louis, MO, USA) and were used without further purification unless otherwise stated. Unmodified nucleoside phosphoramidites and 2-dimethoxytrityloxymethyl-6-(3′,6′-dipivaloylfluorescein-6-yl- carboxamido)-hexyl-1-O-[(2-cyanoethyl)-(*N*,*N*-diisopropyl)]-phosphoramidite were purchased from Glen Research (Davis Drive, Sterling, VA, USA). FESAN was obtained from Katchem (Rež n/Prague). C18 SepPak cartridges were purchased from Waters Corp., (Miliford, MA, USA), and ammonium hydroxide (30%) (J.T. Baker brand) was obtained from Avantor Performance Materials (Center Valley, PA, USA).

Negative ion MALDI mass spectra were recorded on an Axima Performance (Shimadzu, Kyoto, Japan) instrument equipped with a nitrogen laser (337 nm) in the linear mode. A mixture of a 50 mg/mL solution of 3-hydroxypicolinic acid in 50% acetonitrile and a 50 mg/mL solution of diammonium hydrogen citrate in deionized water (8:1, *v/v*) was used as a matrix.

Negative ion electrospray mass spectra (ESI-MS) were recorded on a Synapt G2 Si high-resolution mass spectrometer (Waters) equipped with a quadrupole-time-of-flight mass analyzer (Waters Corp., Miliford, MA, USA). Raw data were collected in the continuum resolution mode and deconvoluted using the MaxEnt1 algorithm to a zero-charge state mass.

All UV absorption measurements were performed in 1 cm path length cells using a GBC Cintra 4040 UV-VIS spectrophotometer (Dandenong, Australia). Solutions of the compounds for UV experiments were prepared by dissolving each compound in deionized water, and the measurements were performed at ambient temperature.

### 4.1. Synthesis of 1,2-Dicarba-Closo-Dodecaborane Tripeds 9,12-Bis-Functionalized with 22-Mer Oligonucleotides (1 and 2) and Tagged with 6-Fluorescein (FL-1 and FL-2)

Synthesis of tripeds **1**, **2**, **FL-1** and **FL-2** was performed according to a previously described procedure [35], by the use of the phosphoramidite solid-phase method, an LCA CPG glass support loaded with triped shaped 1-(2-O-succinylethyl)- 9,12-bis(3-O-tritylprop-1-yl)-1,2-dicarba-*closo*-dodecaborane and commercially available nucleoside phosphoramidites. Synthesis at a 0.1 µmol scale was performed on an H6 GeneWorld automated DNA/RNA synthesizer (K&A, Laborgeraete GbR, Schaafheim, Germany) under the conditions recommended by the manufacturer. After synthesis, 6-FAM-labeled oligonucleotides and tripeds **1** and **2** bearing 5′-DMTr protecting groups were cleaved from the solid support under standard conditions. All the compounds were purified by reverse-phase HPLC and desalted on C18 SepPak cartridges. In the case of tripeds **1** and **2**, subsequent removal of the 5′-DMTr groups was performed on C18 SepPak cartridges by treating the adsorbed oligomers with 2% TFA, followed by washing the cartridges with water (approximately 10 mL) and eluting the desired product with 30–50% CH_3_CN in water (approximately 2.0 mL). The resulting tripeds **1**, **2**, **FL-1**, and **FL-2** were analyzed by ESI-Q-TOF MS and UV-VIS spectroscopy (data shown in Appendix A and in Appendix A). The unmodified oligonucleotides ASO-22, ASO-C, and RNA-1 were synthesized routinely, and their spectral data are given in Appendix A and in Appendix A for ASO-C.

### 4.2. RP-HPLC Analysis

Reverse-phase high-performance liquid chromatography (RP-HPLC) analyses were performed on a Shimadzu Prominence HPLC system (Kyoto, Japan) using a Kinetex 5 µm C-18 column (100 Å, 250 × 4.6 mm column, Phenomenex). All analyses were performed at ambient temperature. The HPLC conditions were as follows: buffer A: 0.1 M CH_3_COONH_4_, pH 6.7/H_2_O; buffer B: CH_3_CN; and flow rate: 1 mL min-1. The gradient of buffer B was as follows: 0→2 min 0%; 2→25 min 0–45%; 25→28 min 45–60%; 28→30 min 60–0%; 30→33 min 0% for **FL-1** (retention time, RT = 14.3 min) and **FL-2** (RT = 13.7 min). UV detection was performed at λmax 260 nm and λmax 494 nm, and the amount of compound was defined in optical units (OD) at the 260 nm wavelength.

### 4.3. Radiolabeling of Tripeds 1 and 2 at Their 5′-Terminal Units

In the experiment, 0.10 OD of triped **1**, 0.12 OD of triped **2**, and 0.1 OD of 22-nt RNA or ASO-22 were dissolved in 15 µL of Milli-Q water and mixed with ^32^P-γ-radiolabeled ATP (37.0 MBq, 1.00 mCi, diluted 10x with Milli-Q water, 2 µL, delivered by Hartmann Analytic, Braunschweig, Germany), T4 polynucleotide kinase (1 µL, 10 unit/mL), and kinase buffer (2 µL) supplied by the manufacturer. The reaction mixtures, which had a total volume of 20 µL, were incubated at 37 °C for 1 h. Next, an enzyme was inactivated by incubation of the reaction mixtures for 3 min at 80 °C and the 5′-radiolabeled products were used in studies without further purification.

### 4.4. Assembly of Tripeds 1 and 2

The formation of DNA nanostructures was initiated by mixing triped **1** or triped **FL-1** (0.10 OD) with triped **2** or triped **FL-2** (0.12 OD), respectively, at a molar ratio of 1:1 in 20 mM Tris-HCl buffer (pH 8) containing 50 mM NaCl and 10 mM MgCl_2_ in a total volume of 10 µL. The mixture was heated for 5 min at 75 °C, slowly cooled (3 h) to room temperature, and then left overnight at 4 °C. The samples were analyzed by nondenaturing (no urea added) 15% polyacrylamide gel electrophoresis (PAGE). A dsDNA marker (GeneRuler Ultra Low Range DNA Ladder, Thermo Scientific, Waltham, MA, USA) was used in a total volume of 20 µL per lane (Figure 2 and Appendix A). Electrophoresis was performed at room temperature at a constant voltage of 300 V/cm and a current of 6 mA for 3 h. The PAGE slab gels for non-radioactively labeled **1**/**2** were stained with Stains-all for 30 min, and then the gels were scanned using a G-Box apparatus (Syngene, Cambridge, UK). For ^32^P-labeled **1**/**2**, the gels were at first radioautographed, and then stained with Stains-All.

### 4.5. Cell Line and Culture Conditions

HeLa (human cervical carcinoma, ATCC, Manassas, VA, USA) cells were cultured in RPMI 1640 medium (Gibco, BRL, Paisley, New York, NY, USA) supplemented with 10% heat-inactivated fetal bovine serum (FBS) (Gibco, BRL, Paisley, New York, NY, USA), 100 U/mL penicillin, and 100 µg/mL streptomycin (Gibco, BRL, Paisley, New York, NY, USA) at 37 °C and 5% CO_2_. The A431 (human squamous carcinoma) and MCF-7 (human breast cancer) cell lines (ATCC, Manassas, VA, USA) were cultured in DMEM containing 4.5 g/L D-glucose, 0.11 g/L sodium pyruvate, and without L-glutamine (Gibco, BRL, Paisley, New York, NY, USA) supplemented with 10% heat-inactivated fetal bovine serum (FBS) (Gibco, BRL, Paisley, New York, NY, USA), 100 U/mL penicillin, and 100 µg/mL streptomycin (Gibco, BRL, Paisley, New York, NY, USA) at 37 °C and 5% CO_2_. Cell culture was analyzed for mycoplasma contamination with the EZ-PCR Mycoplasma Detection Kit (BI, Cromwell, CT, USA) and proved negative (Appendix A). Immediately before transfection, the culture medium was replaced, and the cells were trypsinized (Gibco, trypsin-EDTA (0.5%), no phenol red) and counted.

### 4.6. Stability of Complexes **1**/**2** in A431 Cancer Cell Lysates

A431 cells were cultured as described above. The cells were trypsinized (Gibco, trypsin-EDTA (0.5%), no phenol red) and counted. The cells were washed with 100 µL of cold PBS (with Mg^2+^, Ca^2+^) and lysed in 100 µL of RIPA buffer (50 mM Tris, 150 mM NaCl, 2 mM EDTA, 0.1% SDS, 1% NP-40, pH 7.6) with 1 µL protease inhibitor (100×). After 30 min of incubation on ice, the samples were centrifuged at 14,000 rpm for 15 min at 4 °C, and the protein pellet was collected.

The ^32^P-radiolabeled oligonucleotide triped **1** (0.1 OD, 2 µL of the stock solution after phosphorylation) and triped **2** (0.12 OD, 2 µL of the stock solution after phosphorylation) were mixed at a molar ratio (1:1) in 20 mM Tris-HCl buffer (pH 8) containing 50 mM NaCl and 10 mM MgCl_2_ in a total volume of 15 µL. The mixture was heated for 5 min at 75 °C, slowly cooled (3 h) to room temperature, and left overnight at 4 °C. Oligonucleotide ASO-22 (4 µL of stock solution after phosphorylation) was mixed with 20 mM of Tris-HCl buffer (pH 8) containing 50 mM NaCl and 10 mM MgCl_2_ in a total volume of 15 µL. The samples of **1**/**2** (15 µL) or ASO-22 (15 µL) were mixed with A431 cell lysate (15,000 cells per 1 OD of nucleic acid). The mixture was incubated at 37 °C for up to 48 h. Aliquots of the enzymatic reaction mixture (4 µL) were withdrawn from the reaction mixture at predetermined times (0, 12, 24, 48 h) and mixed with loading buffer (10 mM Tris-HCl, 60% glycerol, 0.03% bromophenol blue, 0.03% xylene cyanol, pH 7.6) (6 µL). Each sample was analyzed by nondenaturing (no urea added) 15% PAGE at room temperature at 20 mA for 2 h. Gels were visualized on radiographic films (Figure 2B,C, right panels). The **1**/**2** and ASO-22 decay courses (Figure 2D) were assessed from the time-dependent intact products content shown in Figure 2B,C (right panels), respectively. Statistical analysis is given in Figure 2’s caption.

### 4.7. In Vitro RNase H—Assisted RNA Cleavage

In the experiment, ^32^P-radiolabeled substrate RNA (2 µL of the stock solution after phosphorylation) was mixed with triped **1** or ASO-22 (0.05 OD, 5 µL) in 20 mM Tris-HCl buffer (pH 8) containing 50 mM NaCl and 10 mM MgCl_2_ in a total volume of 7 µL. The mixture was heated for 5 min at 75 °C and slowly cooled (1 h) to room temperature. To each sample, 10x reaction buffer containing 200 mM Tris-HCl, 500 mM KCl, 200 mM DTT, and 50 mM MgCl_2_ (3 µL) and Milli-Q water (19 µL) were added at 4 °C, and then ribonuclease H (RNase H, Eurx, Poland) (1 µL of 1 U/µL stock solution) was added to a total volume of 30 µL. The resultant assay mixture was incubated at 37 °C for up to 90 min. Aliquots of the enzymatic reaction (3 µL) were withdrawn from the reaction mixture at predetermined times (0, 5, 15, 30, 60, 120, and 240 min) and mixed with loading buffer (10 mM Tris-HCl, 60 mM EDTA, 60% glycerol, 0.03% bromophenol blue, 0.03% xylene cyanol, pH 7.6) (6 µL), and the enzyme was inactivated by incubation of the reaction sample for 3 min at 80 °C. Each sample was analyzed by 20% PAGE with 7 M urea at room temperature at 20 mA for 2.5 h. After electrophoresis was complete, the gels were transferred to an exposure cassette and covered with autoradiography double-coated films (MXBE film, Rochester, NY, USA) for 10 min at low temperature (−25 °C). Then, the double-coated films were soaked in the developing reagent (Kodak Processing Chemicals, Sigma Aldrich, St. Louis, MO, USA) and then in the fixing reagent and scanned using a G-Box apparatus (Syngene, Cambridge, UK). The mixture of ^32^P-isotope labeled oligoribonucleotides (2–22 mers) synthesized *in house* was used as a marker (M) for analysis of data in Figure 3A.

### 4.8. Western Blot Analysis of Downregulation of EGFR in A431 Cells

A431 cells were cultured as described above. Before transfection, the medium was removed, and cells were trypsinized (Gibco, trypsin-EDTA (0.5%), no phenol red) and counted. Then, cells were loaded in 96-well plates with black walls and a transparent bottom (PerkinElmer) at a density of 15 × 10^3^ cells per well in 100 µL of complete medium and cultured for 24 h to reach 80% confluence. Immediately before transfection, the cell medium containing antibiotics was replaced with fresh medium without antibiotics. After 1 h of incubation, A431 cells were transfected with complexes **1**/**2** or ASO-22 dissolved in 50 µL OPTI-MEM medium at concentrations of 0, 5, 25, 50, and 100 nM using Lipofectamine 2000 at a ratio of 2:1 (2 µL of Lipofectamine 2000 per 1 µg of nucleic acid). For control the cells were Lipofectamine 2000-transfected with ASO-C (100 nM). After 48 h of incubation at 37 °C in a 5% CO_2_ atmosphere, cells were washed with cold PBS and lysed with 99 µL of RIPA buffer and with 1% protease inhibitor 100x (1 µL). Samples were incubated for 30 min at 4 °C and centrifuged at 14,000 rpm for 15 min at 4 °C. The protein concentration was determined by the Bradford protein assay (BioRad, Hercules, CA, USA), and 30 µg protein per well was separated on SDS PAGE (4% and 8%) reducing gradient gels (6.8 pH and 8.8 pH, respectively) and transferred to nitrocellulose membranes (Thermo Scientific, Waltham, MA, USA) for semi-dry Western blot analysis. After blocking nonspecific binding sites, membranes were incubated overnight at 4 °C with primary antibodies: anti-actin Mab (1:500) mouse monoclonal antibody, clone C4 (Abnova, Taiwan), and anti-EGFR A10 (1:200) mouse monoclonal antibody (Santa Cruz Biotechnology, Heidelberg, Germany).

Membranes were washed 3 times for 20 min at room temperature before incubation with a secondary antibody for 1 h at 4 °C. The goat anti-mouse IgG (1:5000) (H&L) peroxidase (Abnova, Taiwan) was the secondary antibody. Membranes were scanned using ELC Western Blot Pierce Blotting Substrate (Sigma Aldrich, Oakville, ON, Canada) for 5 min in darkness and analyzed using a G-Box apparatus (Syngene, Cambridge, UK). Experiments were conducted in duplicate and the mean with standard deviation (±SD) is given.

### 4.9. Analysis of the Migration Rate and Cancer Cell Phenotype

HeLa cells were cultured and transfected as described above. After 1 h of incubation, HeLa cells were transfected with Lipofectamine 2000 at a ratio of 2:1 with **1**/**2** or ASO-22 (2 µL of Lipofectamine 2000 per 1 µg of the nucleic acid) used at concentrations of 0, 50, and 100 nM dissolved in 50 µL OPTI-MEM medium. For control the cells were Lipofectamine 2000-transfected with ASO-C (100 nM). For the wound-healing assays, after 5 h of transfection at 37 °C in an atmosphere of 5% CO_2_, cells in 96-well plates were wounded by scraping with pipette tips (of 20 µL size). The cells were washed two times with a medium, and the wound areas were marked and photographed at different points using a digital camera attached to a phase-contrast microscope (Nikon-Eclipse, Tokyo, Japan). The number of cells in the wound region was counted. Experiments were conducted in triplicate and the mean with standard deviation (±SD) are given.

After 48 h of incubation, HeLa cells in each well were photographed at the same points for phenotypic change assays and wound healing assays. Cells in the wound areas were counted under a bright-field microscope at a magnification of 200× across 4 randomly selected fields.

### 4.10. Cell Cycle Analysis

A431 cells were cultured and transfected with nanostructures **1**/**2** or ASO-22 as described above. A431 cells treated with Lipofectamine 2000 at the concentration corresponding to transfection of **1**/**2** at 100 nM were used as a control. After 48 h of incubation at 37 °C in an atmosphere of 5% CO_2_, under the same conditions, cells were harvested and transferred to centrifuge tubes and spun for 5 min at 1100 rpm. Each cell sample was then suspended in cold 70% ethanol for fixation for 1 h. Then, the cells were centrifuged for 10 min at 1500 rpm and 4 °C and washed twice in PBS without Ca^2+^ and Mg^2+^ ions. Next, cells were treated with 10 µL of RNase A (10 mg/mL) for 30 min at 36 °C to ensure that only DNA, and not RNA, was stained. In the next step, 5 µL of propidium iodide (1 mg/mL) was added, and the cells were incubated at room temperature for 30 min. Each cell sample was then transferred to a 5 mL polystyrene round-bottom tube (12 × 75 mm style, Falcon). Flow cytometry and corresponding fluorescence measurements were performed using a BD FACSCalibur flow cytometer (Becton-Dickinson, East Rutherford, NJ, USA) using the FL2 Red channel. The number of events was stopped at 25,000 counts. Data collected from the experiments were analyzed using ModFit LT software. FACS analysis with confidence interval (CI) and the mean with standard deviation (±SD) statistics is given in Appendix A.

### 4.11. Microscopic Analysis of the FL-1/FL-2 Localization in MCF-7, HeLa and A431 Cells

MCF-7, HeLa, and A431 cells were cultured and transfected as described above. After 5 h of incubation, the transfection mixture was replaced with a full fresh medium with antibiotics in a volume of 200 µL per well. Next, after 6 h of incubation, the full medium was mixed with 5 µg/mL DAPI dye (which stains cell nuclei blue) (Invitrogen, Thermo Fisher Scientific, Waltham, MA, USA) and 1 µmol/mL ER-Tracker Red dye (which stains ER membranes red) (Invitrogen, Thermo Fisher Scientific, Waltham, MA, USA). The cells were incubated for 30 min with the dyes in the dark at 37 °C. After incubation, the cells were washed two times with PBS buffer (with Ca^2+^ and Mg^2+^) and imaged under a fluorescence microscope (Nikon-Eclipse, Tokyo, Japan) to detect DAPI at λ_ex_ = 340–380 nm (λ_DM_ = 400 nm, λ_BA_ = 435–485 nm) and FITC at λ_ex_ = 465–495 nm (λ_DM_ = 505 nm, λ_BA_ = 515–555 nm), B-2A (longpass, λ_ex_ = 450–490 nm(λ_DM_ = 505 nm, λ_BA_ = 520 nm), TX red (λ_ex_ = 540–580 nm, λ_DM_ = 595 nm, λ_BA_ = 600–660 nm), and G-2A longpass λ_ex_ = 510–560 nm (λ_DM_ = 575 nm, λ_BA_ = 590 nm), where DM is a dichroic mirror and BA is an emission filter.

### 4.12. Human Primary Monocyte-Derived Macrophages and Measurement of Activation of the NLRP3 Inflammasome

Peripheral blood mononuclear cells were isolated from leukocyte-rich buffy coats provided commercially by blood transfusion and donation center (the Regional Blood Transfusion Centre in Lodz, Poland). The blood donors agreed on donating the buffy coat randomly for research purposes during blood donation, maintaining their anonymity. Monocytes selected by adhesion to plastic were differentiated into macrophages in macrophage-SFM medium (Gibco, Thermo Fisher Scientific, Waltham, MA, USA) supplemented with 10 ng/mL granulocyte-macrophage colony-stimulating factor (GM-CSF, Immunotools GmbH, Friesoythe, Germany) and antibiotics as described previously [71,72]. On day six, the cells were washed two times with PBS, supplied with fresh RPMI 1640 medium (without FBS) supplemented with L-glutamine and antibiotics, and primed with 1 μg/mL lipopolysaccharide (LPS) (InvivoGen, San Diego, CA, USA) for 4 h to induce the expression of nucleotide-binding oligomerization domain and leucine-rich repeat-containing receptor family pyrin domain containing 3 (NLRP3) inflammasome components and IL-1β. Subsequently, the cells were treated with boron clusters (1,2-DCDDB, FESAN), **1**/**2** or ASO-22 (50 nM) for 16 h or with control nigericin (InvivoGen, 20 μM) for 4 h to activate the NLRP3 inflammasome. The level of the released IL-1β was measured from conditioned macrophage media by enzyme-linked immunosorbent assay (ELISA, R&D Systems, Minneapolis, MN, USA) according to the manufacturer’s instructions. Experiments were conducted in two independent donors.

### 4.13. Determination of Cytotoxicity of 1/2, ASO-22 as well as 1,2-DCDDB and FESAN in Human Macrophages (MTT Assay)

The cytotoxicity of boron clusters (1,2-DCDDB, FESAN) as well as **1**/**2** and ASO-22 (50 nM) in macrophages was assessed with the use of the MTT assay. The macrophages were cultured as described above. The cells were incubated for 16 h with tested compounds at 37 °C under a 5% CO_2_ atmosphere, followed by the addition of the MTT solution in PBS (5 mg/mL) to each well. The cells were then incubated for 3 h at 37 °C under a 5% CO_2_ atmosphere. Finally, 95 µL of lysis buffer (NP-40, 20% SDS, 50% aqueous dimethylformamide, pH 4.5) was added to each well and cells were incubated overnight at 37 °C. The sample absorbance was measured at two wavelengths: 570 nm and the reference wavelength of 630 nm (colorless walls plate reader, PerkinElmer, Waltham, MA, USA). The percentage of living cells (%LC) was calculated from the equation: %LC = Asample × 100%/(Acontrol cells) where Asample is the absorbance of a given sample of cells treated/transfected with the tested compound and Acontrol cells is the absorbance of the untreated cells (100% viability). The results are mean values ±SD from two technical experiments.

### 4.14. Determination of the Boron Content in A431 Cells by ICP MS Measurements

A431 cells were cultured as described above and then transfected with aqueous solutions of **1**/**2** (0.1, 0.4, 0.8, 2, and 4 µM) and Lipofectamine 2000 at a ratio of 1:2 dissolved in 50 µL OPTI-MEM medium. After 5 h of incubation, the transfection mixture was replaced with a fresh medium containing antibiotics in 200 µL per well and the cells were incubated for 48 h at 37 °C in an atmosphere of 5% CO_2_. In order to conduct the experiment for the assessment of the free uptake of **1**/**2**, the medium was replaced to basic medium (DMEM containing 4.5 g/L D-glucose, 0.11 g/L sodium pyruvate, and without L-glutamine) (Gibco, BRL, Paisley, New York, NY, USA), 100 U/mL penicillin, and 100 µg/mL streptomycin (Gibco, BRL, Paisley, New York, NY, USA) and the aqueous solutions of **1**/**2** (0.1, 0.4, 0.8, 2, and 4 µM) were added to the medium and the cells were incubated for 48 h at 37 °C in an atmosphere of 5% CO_2_.

After 48 h (experiments with or without Lipofectamine 2000) and medium removal, the cells were treated with trypsin (1.5 h, 20 µL/well, Gibco, trypsin-EDTA, 0.5%, no phenol red) and diluted with ultrapure water (18 MΩ·cm, Simplicity Water Purification Systems, Millipore SAS, Molsheim, France) up to a 1.5 mL volume; then, the ICP MS measurements were performed and data were collected. The boron content was determined with a Varian 810-MS inductively coupled plasma mass spectrometer (Varian, Palo Alto, CA, USA) equipped with a Micromist nebulizer, quartz Scott spray chamber (3 °C), platinum sampler cone, and nickel skimmer cone. The operating parameters were as follows: RF power: 1.4 kW; plasma flow (argon): 17 L/min; auxiliary flow (argon): 1.7 L/min; nebulizer flow (argon): 1.00 L/min; pump rate: 4 rpm; sheath gas (argon): 0.2 L/min; and number of scans: 10 m/z: 10 and 11.

The calibration curve method was applied. A series of 10 calibration solutions were prepared in the boron concentration range of 0.1–50 µg/L by appropriate dilution of the standard solution of boron (1000 mg/L, Merck, Darmstadt, Germany) using ultrapure water. A linear model of the calibration curve was adopted, with a minimum correlation coefficient of 0.999. The obtained results were the average of concentrations calculated from the calibration curves obtained for both stable boron isotopes (^10^B and ^11^B). Six independent experiments were performed.

### 4.15. Statistical Analysis

The statistical analysis was carried out for the results shown at Figure 2D, Figure 3C,E, Figure 5C and Figure 8. At least three independent experiments were performed, unless otherwise stated. The differences between two independent groups of data were calculated by the use of the parametric test, Student’s t-test. To compare the means from three or more groups, the one-way ANOVA and post-hoc Tukey HSD test (indicating an overall statistically significant difference in the group means) with post-hoc Tukey honestly significant difference was used. The HSD (honestly significant difference) test was conducted to confirm where the differences occurred between groups. The homogeneity of variances was verified by Levene’s test and Brown–Forsythe test. All statistical analyses were carried out using Statistica ver. 8.0 software (StatSoft Inc., Tulsa, OK, USA). A value of *p* < 0.05 was considered to be statistically significant.

## 5. Conclusions

Nanostructures built of tripeds **1** and **2** exhibit the ability to regulate EGFR target gene expression and reduce cell migration rate by inhibiting the cell cycle. Notably, the use of boron cluster-DNA conjugates does not exert inflammasome activation in human macrophages, in contrast to metallacarborane cages. A slightly increased nucleolytic stability of the **1**/**2** nanostructures compared to the nonmodified antisense oligonucleotide leaves some room for further structural optimization of the obtained nanoparticles, mostly by insertion of nuclease-resistant modifications, such as locked nucleic acids (LNAs) or 2′-OMe at their 5′-ends. Moreover, the attachment of additional boron clusters and their enrichment with ^10^B isotope atoms may allow the **1**/**2** nanocarriers to achieve a ^10^B concentration sufficient for successful BNCT. Thus, the favorable properties of **1**/**2** nanostructures, both in terms of their proven antisense properties and their potency to serve as carriers of boron atoms for BNCT, make these constructs of interest for further development as anticancer agents.

## Figures and Tables

**Figure 1 ijms-22-04863-f001:**
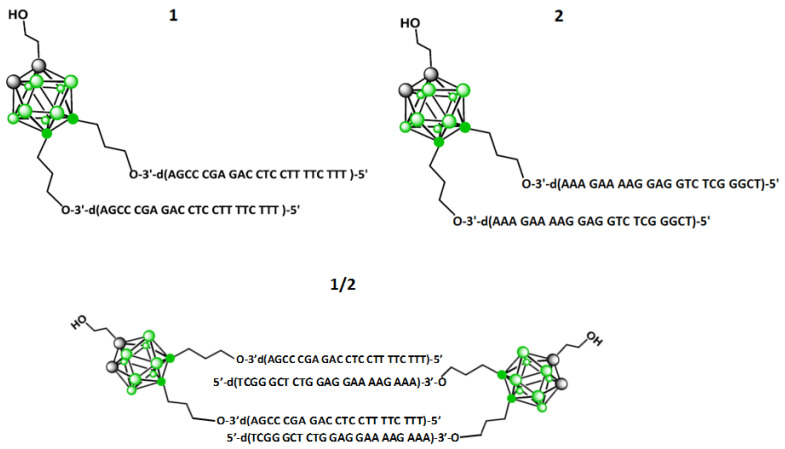
Chemical structures of tripeds **1** and **2**, containing the 1,2-dicarba-*closo*-dodecaborane scaffold *bis*-functionalized with either 22 nt 3′→5′-DNA oligonucleotides complementary to the coding fragment of mRNA of epidermal growth factor receptor (EGFR) or with homologous 22 nt oligonucleotides, respectively, and the chemical structure of dimer **1**/**2**, composed of one molecule of triped **1** and one molecule of triped **2**, of total 44 bp.

**Figure 2 ijms-22-04863-f002:**
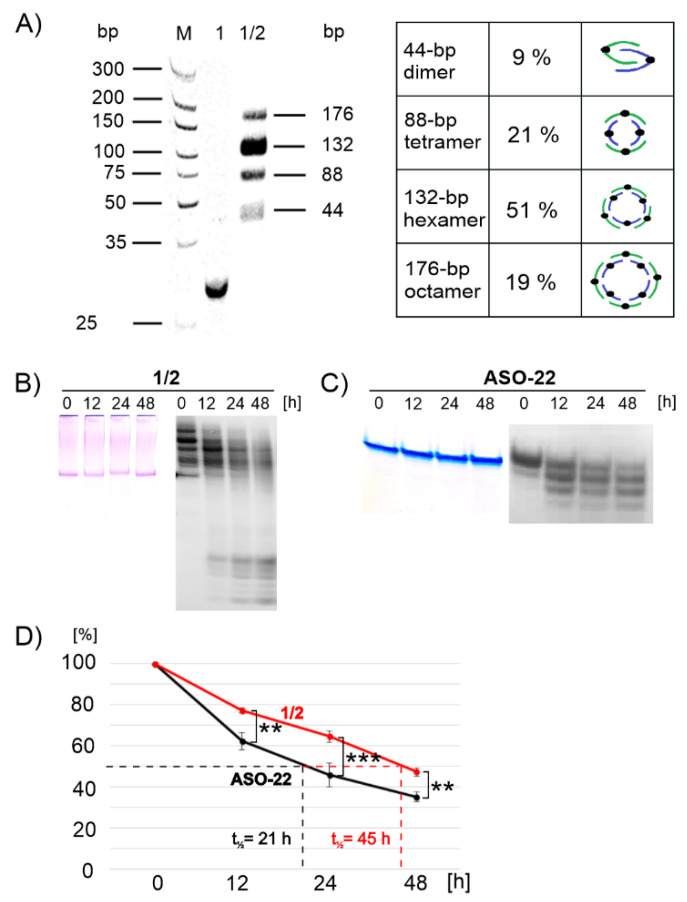
Analysis of the assembly (**A**) and stability of nanostructures **1**/**2** (**B**,**D**) in comparison to the stability of the reference oligonucleotide ASO-22 (**C**,**D**) in DMEM cell culture medium (left panels of (**B**) and (**C**)), and in A431 cell lysate (right panels of (**B**) and (**C**)) at 37 °C. Electrophoretic analysis of the assembly of oligopeds **1** and **2** annealed at a 1:1 molar ratio was performed using a nondenaturing 15% polyacrylamide gel and visualized by Stains-All staining ((**A**), lane of dsDNA size marker–M, and (**B**) and (**C**), left panels) or by autoradiography ((**A**), and lanes for **1** and products of assembly of tripeds **1** and **2**, and (**B**) and (**C**), right panels). The plot of the data in the right panels of (**B**) and (**C**) (degradation of the reference ASO-22 and **1**/**2** assembly products) is shown in (**D**). ** *p* ≤ 0.01, *** *p* ≤ 0.001 (Student’s t-test).

**Figure 3 ijms-22-04863-f003:**
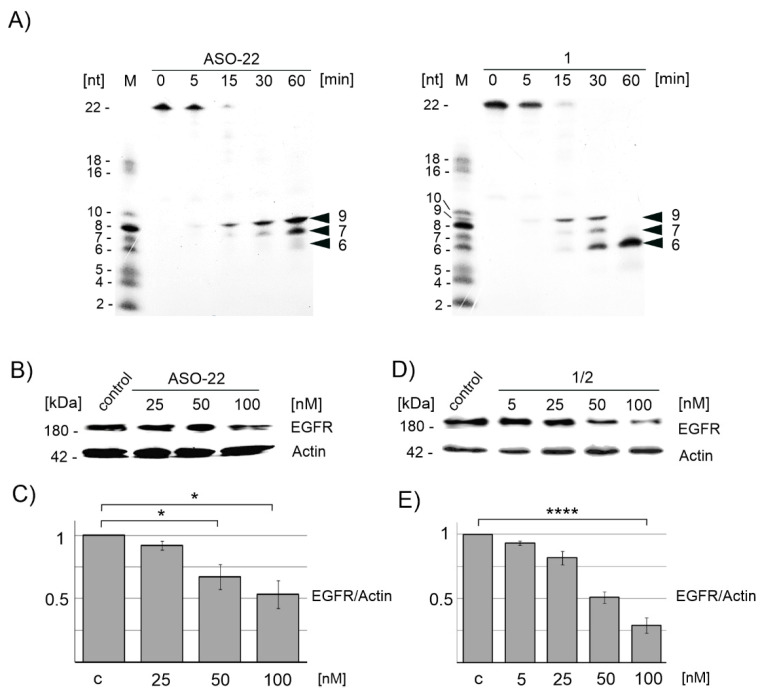
Cleavage of the [^32^P]-RNA target fragment of the EGFR gene hybridized with triped **1** and with ASO-22 monitored electrophoretically (PAGE) under in vitro conditions in the presence of recombinant RNase H (**A**); silencing activity of ASO-22 and **1**/**2** towards the endogenous mRNA of EGFR protein in A431 cancer cells analyzed by immunoblot imaging ((**B**,**D**), respectively) and quantified in the plot (**C**) * *p* ≤ 0.05 and (**E**) **** *p* ≤ 0.0001 (ANOVA and post-hoc Tukey HSD test), respectively. A431 cells transfected with ASO-C are the control and M is the ^32^P-isotope labeled mixture of oligoribonucleotides (2–22 mers).

**Figure 4 ijms-22-04863-f004:**
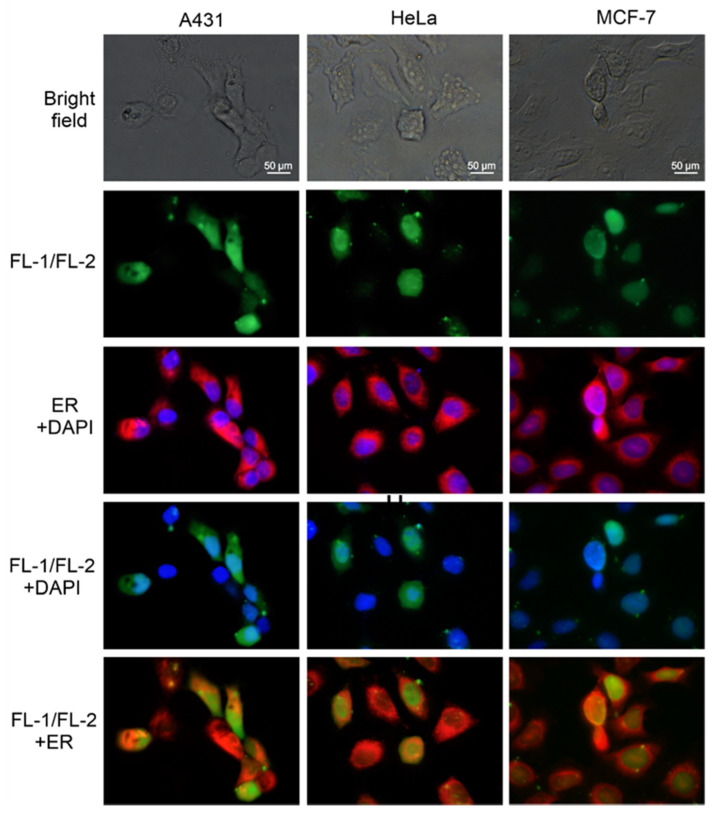
Localization of the **FL-1/FL-2** nanostructures in A431, HeLa, and MCF-7 cancer cells (6 h after transfection). Visualization of green fluorescent B-ASO spots and merged blue cell nuclei (DAPI) and red ER membranes (ER-Tracker Red) are shown in rows 2 and 3, respectively. Merged images of the **FL-1/FL-2** nanostructures either with cell nuclei or with ER are shown in rows 4 and 5, respectively. All panels were enlarged 60 times.

**Figure 5 ijms-22-04863-f005:**
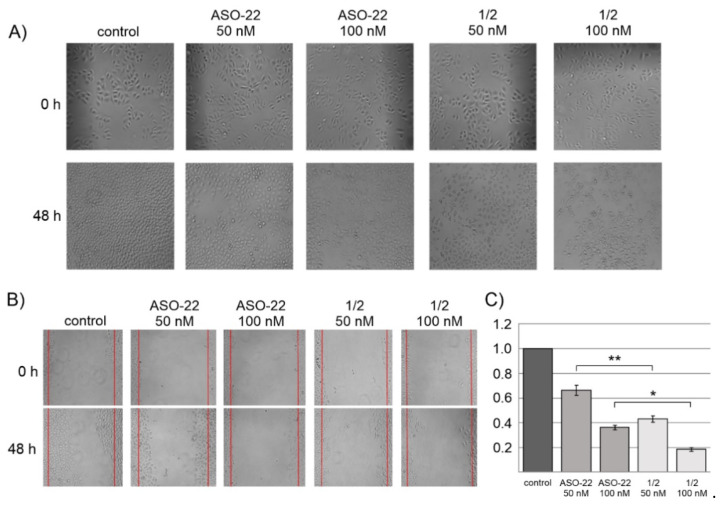
Influence of ASO-22 and nanostructures **1**/**2** on the HeLa cell phenotype (**A**) and migration rate (wound healing assay, (**B**)) and graphical plot of the wound healing assay (**C**). HeLa cells were transfected with ASO-C (100 nM) (nonsense control) or the test compounds (0, 50, 100 nM) in the presence of Lipofectamine 2000. After 48 h of incubation, phenotypic/migration changes of live HeLa cells were microscopically analyzed. All the panels were enlarged 40 times. **** *p* ≤ 0.0001 (ANOVA and post-hoc Tukey HSD test, * *p* ≤ 0.05, ** *p* ≤ 0.01)).

**Figure 6 ijms-22-04863-f006:**
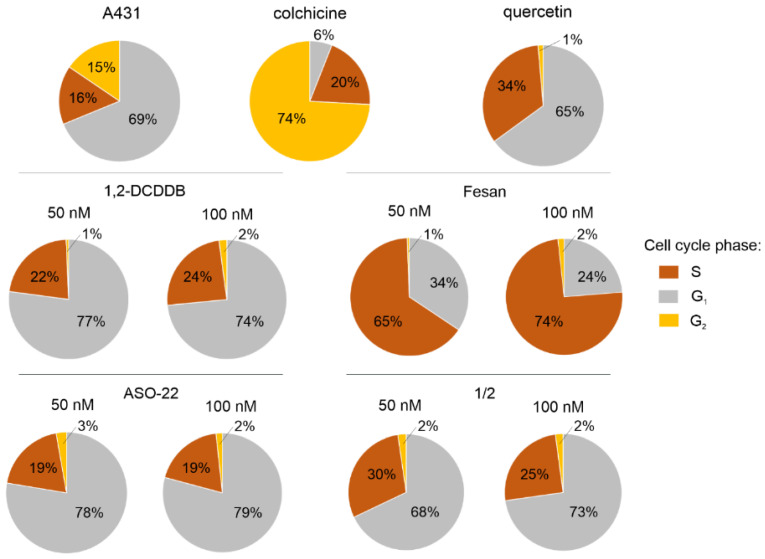
FACS analysis of the cell cycle phases of A431 cells upon Lipofectamine 2000-assisted transfection with ASO-22 and nanostructures **1**/**2** (50 and 100 nM) after 48 h of incubation, and for A431 cells treated with 1,2-DCDDB and FESAN (50 and 100 nM). The A431 cells treated with Lipofectamine 2000 only were used as a control. Colchicine and quercetin were used as positive controls.

**Figure 7 ijms-22-04863-f007:**
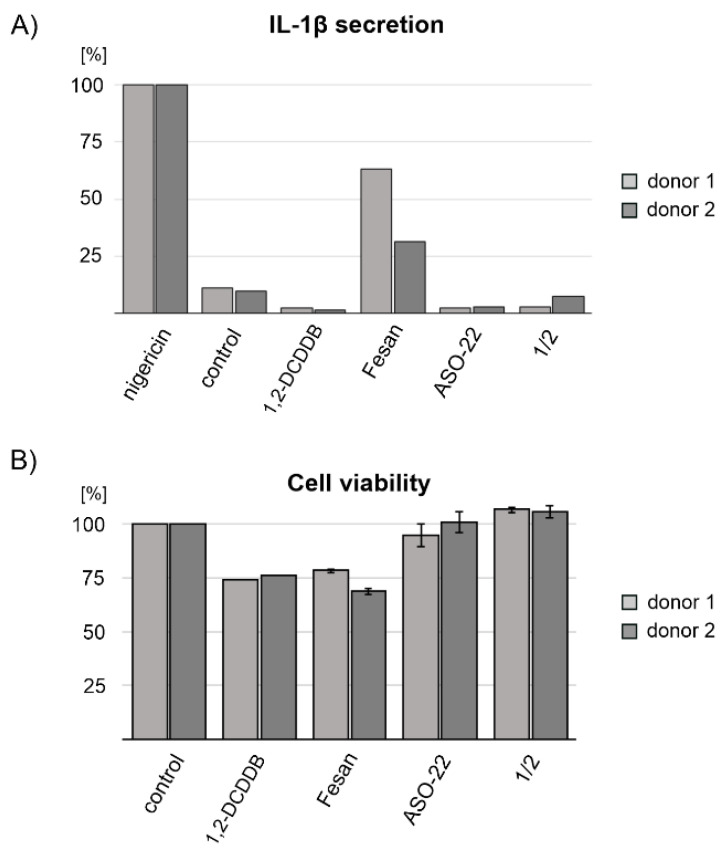
Secreted IL-1β levels (**A**) and cell viability assay (**B**) of human primary monocyte-derived macrophages obtained from two donors (1 and 2) upon transfection with nanostructures **1**/**2** or oligonucleotide ASO-22 or treatment with boron clusters 1,2-dicarba-*closo*-dodecaborane (1,2-DCDDB) and FESAN. For IL-1β ELISA, LPS-primed human primary monocyte-derived macrophages were stimulated with the tested compounds at a concentration of 1 μg/mL. Non-stimulated cells were used as a control. As a positive control for IL-1β release, cells were stimulated with nigericin, and the level of IL-1β secretion for the test compounds was expressed as that of nigericin-induced IL-1β. The viability of cells was determined by the MTT assay. ±SD is given for two technical repeats.

**Figure 8 ijms-22-04863-f008:**
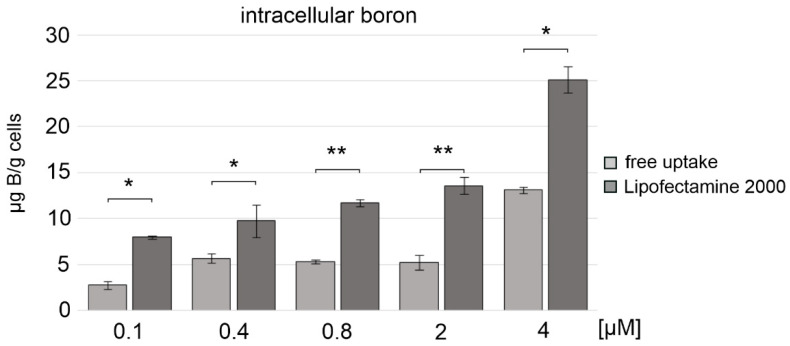
Content of boron atoms in cell lysates of A431 cells transfected with nanostructures **1**/**2** in the presence of Lipofectamine 2000 or delivered by free uptake (no Lipofectamine 2000 used), as measured by the ICP MS method. Nanostructures **1**/**2** delivery with Lipofectamine 2000 *** *p* ≤ 0.001 (ANOVA and post-hoc Tukey HSD test) and without lipofectamine **** *p* ≤ 0.0001 (ANOVA and post-hoc Tukey HSD test). Comparison of the results of boron delivery by **1**/**2** with and without Lipofectamine at a given concentration was calculated * *p* ≤ 0.05, ** *p* ≤ 0.01 (ANOVA and post-hoc Tukey HSD test).

## Data Availability

Available under request.

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
