# Peer review of "Composites of Nucleic Acids and Boron Clusters (C2B10H12) as Functional Nanoparticles for Downregulation of EGFR Oncogene in Cancer Cells"

_ijms, 2021, doi:10.3390/ijms22094863_

Round 1
Reviewer 1 Report
The manuscript «Composites of nucleic acids and boron clusters (C2B10H12) as functional nanoparticles for downregulation of EGFR oncogene in cancer cells» submitted by Damian Kaniowski et al. is devoted to deepened studies of functional DNA constructs conjugated with boron clusters. Authors confirmed their earlier results concerning the structures and sizes of B-ASO formed by two complementary oligonucleotides constructions presented by 1,2-dicarba-closo-dodecaborane linked with two oligonucleotides. A slightly increased nucleolytic stability of the nanostructures compared to the nonmodified antisense oligonucleotide and their ability to regulate EGFR target gene expression and reduce cell proliferation by inhibiting the cell cycle were demonstrated. These nanoconstructions have the potential to exert a dual anticancer effect: antisense EGFR downregulation and delivery of boron atoms for boron neutron capture therapy (BNCT). The complex of modern methods of the investigation of the mechanisms of action of B-ASO was used in present work, such as, plasma mass spectrometry (ICP MS), FACS analysis, dual fluorescence assay (DFA) microscopy, Western blot analysis, Flow cytometry, fluorescence microscopy and others. I recommend accepting this manuscript for publication minor revision.
I have some questions and remarks:
- Figure 2. Why did you use Stains-all staining when you investigated stability in DMEM and radioactive label for cell lysate? How can you enplane the absence of nanostructures in DMEM in comparison with cell lysate?
- Did you observe two products upon 32P-labelling of 1 and 2 due to possibility of introduction of one or two 5'-phosphates in your construction?
- Figure 3. What does mean M in panel 3A. Please describe in Materials and methods how you obtained this type of marker.
- The triped 1 can formed two duplexes with complementary RNA. How do you think are there any possibility to simultaneously binding of two RNase H molecules with such complex. Can this process affect to the effectiveness and site specificity of RNase H activity?
- I think it is necessary to add the sequence of ASO-C in supplementary table S1. How did you chose this sequence?
- Have you the agreements from two donors for using of their biological material for scientific investigations? I'm not sure, but It is possible that the agreements from these persons is necessary for publication. Can you clarify this question?
- Figure 7. I recommend to change minus "-" in diagram capture for "control" or something else due to possible misunderstanding.
- Line 565. Correct the number of subsection and "5'-Radiolabeling…. ".
- Line 642. The point between two sentences is missed.
- Line 640. Do you use cyanol or xylene cyanol?
- Line 651. What pH the buffer had?
- Line 660. What instruments did you use for radiographic films visualization and processing?
- Lines 795-801. The regular font for 1, 2, ½ at el. can be changed for bold.
- Line 868. Please change ":" for "," between volume number and pages.
Author Response
Reviewer #1
Dear Reviewer! Thank you very much for all the critical comments. We did our best to
address your doubts and critics and improve the manuscript. Below are the answers for each
point.
1. Figure 2. Why did you use Stains-all staining when you investigated stability in DMEM and
radioactive label for cell lysate?
In the first step, we examined the stability of the oligonucleotide models in the cell medium
and in cell lysates by staining the gels with Stains-all, which is a routine method for staining
the electrophoretic gels. Stability analysis in DMEM indicated complete resistance of
oligonucleotides in the screened conditions. On the other hand, oligonucleotide stability in
cell lysates was difficult to assess when using Stains-all, due to the fact that cell lysates are
rich in enzymatic and structural proteins, lipids and cell fractions in contrast to the basic cell
medium and no clear picture of the oligonucleotide hydrolysis could be obtained (smears).
Therefore, we decided to perform digestion analysis in lysates after radioactive labeling of
oligos. In the meantime, it turned out that the labeling of the oligonucleotides greatly
facilitated the visualization of the complexes formed from triped 1 and 2 and allowed to
assess the stability of 1/2 in cell lysates in a more clear way.
2. How can you explain the absence of nanostructures in DMEM in comparison with cell
lysate?
In the analysis in DMEM medium, we could see the dimer complex and the smear of the
higher molecular weight products of lower mobility, albeit this smear was not separated into
individual complexes. In contrast, after attachment of the 32P-labeled phosphate group to the
5’-end of oligonucleotides the separation of bands for individual complexes was very
efficient. We have no rational explanation for this phenomenon.
3. Did you observe two products upon 32P-labelling of 1 and 2 due to possibility of
introduction of one or two 5'-phosphates in your construction?
After labeling of the triped 1 or 2, we observed one product only on the PAGE gel, what may
indicate that phosphate groups are attached to both oligos in each triped. However, the
electrophoretic mobility of the single-labeled triped should be almost the same as doublelabeled triped taking into account that attachment of one or two phosphate groups results in
(i) increase of MW by 1.16% (for PO3 x 2) or 0.58% (for PO3 x1) and (ii) additional two or
four negative charges over 42 negative charges of the triped molecule.
4. Figure 3. What does mean M in panel 3A. Please describe in Materials and methods how
you obtained this type of marker.
M - is a description of the marker of RNA oligos of the length of 22,18,16,10,9,8,7,6,5,4 and 2
nt. This marker mixture was prepared in house and labelled radioactively with 32P for the
described use. The proper text extension is added in Material & Methods section and in the
caption under Figure 3.
Figure 3: A431 cells transfected with ASO-C are the control and M is the 32P-isotope
labeled mixture of oligoribonucleotides (2-22 mers).
Materials & Methods: The mixture of 32P-isotope labeled oligoribonucleotides (2-22 mers)
synthesized in house was used as a marker (M) for analysis of data at Fig. 3A.
.
4. The triped 1 can formed two duplexes with complementary RNA. How do you think are
there any possibility to simultaneously binding of two RNase H molecules with such complex.
Can this process affect to the effectiveness and site specificity of RNase H activity?
We believe that such a case can occur in vitro, where two RNA molecules associate to one
triped molecule (1), and it is conceivable that then two protein molecules process both
duplexes simultaneously. The obtained results present in Fig. 3 suggest that the rate of
hydrolysis of 1 is slightly higher than that of the ASO, what is in accordance with what we
think, although no direct evidence was obtained. Under cellular conditions, however, the
activity of 1/2 is similar to that of ASO-22, which may indicate that one triped molecule binds
to one mRNA molecule and activates 1 enzyme molecule only.
5. I think it is necessary to add the sequence of ASO-C in supplementary table S1. How did
you chose this sequence?
The ASO-C sequence was selected by us in our previous studies as a molecule which does
not change the cell metabolism (no complementarity to genes important for cell metabolism
verified by the NCBI BLAST sequence analyzer) (as the guide strand of the siRNA sequence
in Sipa K, Sochacka E, Kazmierczak-Baranska J, Maszewska M, Janicka M, Nowak G,
Nawrot B. Effect of base modifications on structure, thermodynamic stability, and gene
silencing activity of short interfering RNA. RNA. 2007 Aug;13(8):1301-16 and as nonsenseASO published in ref. 44 and 45 given in the text). The sequence of ASO-C is inserted into
Table S1, and the info on ASO-C is given in section 2.4.3 of the main text (ref[44,45])..
6. Have you the agreements from two donors for using of their biological material for
scientific investigations? I'm not sure, but It is possible that the agreements from these
persons is necessary for publication. Can you clarify this question?
Primary monocyte-derived macrophages are isolated routinely from buffy coats provided
commercially by blood transfusion and donation center. The blood donors agreed on
donating the buffy coat randomly for research purposes during blood donation, maintaining
their anonymity. Therefore this material cannot be treated as clinical/patient material as
understood in clinical studies.
Therefore we modified Materials & Methods section 4.12 as follow:
Peripheral blood mononuclear cells were isolated from leukocyte-rich buffy coats provided
commercially by blood transfusion and donation center obtained from the Regional Blood
Transfusion Centre in Lodz, Poland). The blood donors agreed on donating the buffy coat
randomly for research purposes during blood donation, maintaining their anonymity.
7. Figure 7. I recommend to change minus "-" in diagram capture for "control" or something
else due to possible misunderstanding.
The respective correction was done in figure 7 and in the caption of this figure.
8. Line 565. Correct the number of subsection and "5'-Radiolabeling…. ".
The title has been changed to: “4.3. Radiolabeling of tripeds 1 and 2 at their 5’-terminal units”
9. Line 642. The point between two sentences is missed.
The sentences were reworded accordingly.
10. Line 640. Do you use t or xylene cyanol?
The respective correction (xylene cyanol) was done in lines 642 and 659 of the main text.
11. Line 651. What pH the buffer had?
The pH value (8.0) was originally given in the text (“….20 mM Tris-HCl buffer (pH 8) 650
containing 50 mM NaCl…) in line 650.
12. Line 660. What instruments did you use for radiographic films visualization and
processing?
Autoradiography double-coated films were purchased from Carestream Medical X-ray blue
(MXBE film, Rochester, NY, USA). The developer reagent and fixer reagent were from
Kodak Processing Chemicals (Sigma Aldrich, St. Louis, MO, USA). After electrophoresis was
complete, the gel was transferred to an exposure cassette and covered with autoradiography
double-coated films for 10 min at low temperature (−25 °C). Then, the double-coated film
was soaked in the developing reagent and then in the fixing reagent and scanned using a GBox apparatus (Syngene, Cambridge, UK).
The following text was added to paragraph 4.7 : After electrophoresis was complete, the gels
were transferred to an exposure cassette and covered with autoradiography double-coated
films (MXBE film, Rochester, NY, USA) for 10 min at low temperature (−25 °C). Then, the
double-coated films were soaked in the developing reagent (Kodak Processing Chemicals,
Sigma Aldrich, St. Louis, MO, USA) and then in the fixing reagent and scanned using a GBox apparatus (Syngene, Cambridge, UK).
13. Lines 795-801. The regular font for 1, 2, 1/2 at el. can be changed for bold.
All the requested corrections (the numbers of tripeds given in bold) were done in the main
text in the paragraph on the Supplementary Materials content.
14. Line 868. Please change ":" for "," between volume number and pages.
This correction has been done accordingly, in Ref. 20.

Reviewer 2 Report
The authors introduce very novel binary approach for cancer therapy to knockdown the EGFR expression and deliver the 10 B atoms for BNCT.
They clearly demonstrated this antisense oligonucleotides were stable and apparently suppress the EGFR expression in vitro for some cancer cells.
However this reviewer has the critical criticisms for the authors.
How do the authors introduce this antisense oligonucleotides in vivo especially for human patients? At least the authors should show the bio distribution analysis using this oligonucleotides in in vivo tumor models.
If there is some evidence, this manuscript should be acceptable.
Author Response
Reviewer # 2
How do the authors introduce this antisense oligonucleotides in vivo especially for human patients? At least the authors should show the bio distribution analysis using this oligonucleotides in in vivo tumor models.
Thank you very much for your constructive comment!
In our article we emphasize that we focus on the basic research to answer the question whether the previously described nanostructures containing functional antisense oligonucleotides targeting the EGFR gene demonstrate the desired biological properties in the cellular system. Hence our research on both assembly and stability under cellular conditions, as well as basic research on the impact of nanostructures on the cell cycle and
the silencing potential of these constructs at the protein level is discussed. Besides, we demonstrate that nanostructures 1/2 are boron atom vehicles in a free uptake for A431 cancer cells. These studies constitute a proof of concept for the use of nanostructures as inhibitors of the expression of target genes in cellular system and open the way for further studies.
The next step of our studies will be to create nucleolytically stabilized nanostructures, equipped with functional groups targeting and delivering the B-ASOs to a specific type of cancer cells and tissues, not harmful to normal cells. We are aware that lipofectamine is not feasible as mean of delivery of antisense oligonucleotides in vivo. Oligonucleotide drugs are administered in vivo decorated with targeting or transporting molecules (cholesterol, CPP,
PEG, GalNac etc). We have to check whether any additional molecules introduced to our tripeds do not disturb nanostructures assembly. We are currently implementing a project in which we test the suitability of our decorated nanostructures for BNCT therapy towards in
vivo system. Thus, we still have a few validation studies in simpler model systems before we begin our mouse studies. In the future, we will conduct ADME research in the animal model (cancer tissue xenografts).

Reviewer 3 Report
The aim of the present manuscript was to investigate the possible use of boron clusters linked with anti-EGFR antisense oligonucleotides as functional nanoparticles in antitumor therapy. The approach to the work is interesting, the authors tackle diverse aspects of relevance to characterize these compounds, implementing different techniques. Overall, the results provide in this manuscript are consistent although some points should be addressed to greatly strengthen its quality.
General comment:
How is the selectivity of these complexes? Have they been tested in normal cells? It should be interesting to study their free uptake and accumulation in normal cells and compare it with that in tumor cells.
Specific comments:
- Figure 2A: 88-bp is not trimer but tetramer
- Figure 3B,C: The silencing effect of unmodified ASO-22 on the endogenous EGFR mRNA should be analysed as a reference (as it has been done in experiments shown in figures 3A an D).
- The interpretation of microscopy images of the intracellular localization of nanostructures is complex (Fig. 4). The authors state that they localize predominantly in the cytoplasm. However, fluorescent nanostructures are also clearly observed in the nucleus. By means of images shown in figure 4, it is difficult to ascertain which the main localization is. Quantification of the images could be helpful to clarify that point.
- Most of the studies have been performed in A431 cells which, according to the authors, express high levels of EGFR. Why do they use Hela cells for proliferation/migration assays shown in figure 5? It should be more rational to do this study also in A431 cells.
Images in Figure 5A show changes in morphology as well as in the number of cells 48 h after transfection, but they do not provide enough evidences to affirm: “More significant phenotypic changes were observed in cells transfected with 1/2 than in cells transfected with ASO-22 (Figure 5A). This result may indicate a higher intracellular activity of anti-EGFR antisense oligonucleotides present in the 1/2 nanostructures than in the case of free ASO-22.” (page 8, lines 284-287). This sentence should be modified.
On the other hand, the wound healing assay is a cell migration assay and no conclusions about inhibition of cell proliferation must be drawn from it. There are other methods more precise and quantitative to study cell proliferation. It would also be interesting to analyse apoptosis / cell death in transfected cells.
- Authors state that “the presence of 1,2-DCDDB in nanostructures 1/2 enhanced the antiproliferative properties of the anti-EGFR antisense oligonucleotides”, while explaining the results shown in Figure 6. As transfection with nanostructures 1/2 as well as with ASO-22 inhibit the progression to the G2 phase (2% in both cases), it would be more accurate to say that both have a similar antiproliferative ability but arrest the cell cycle in a different way (different ratio of cells in the S and in the G1 phase).
How many cell cycle experiments were performed? Is there any statistical significance in these results?
- MTT assay is not described in the Materials and Methods section. How long were the cells incubated for the MTT assay? How do you explain that the boron cluster 1,2-DCDDB affects macrophages viability while there is no effect of nanostructures?
- Authors analyse IL-1ß production by macrophages incubated with the nanostructures to rule out an inflammatory response. Although this is a marker of inflammation, it is not the only one. I would suggest to replace the term “inflammatory/inflammation” throughout the manuscript with a more appropriate specific term.
Similarly, the term “immunogenicity” in line 510 (page 14) should be eliminated, as it has not been extensively studied.
- References 51 and 52 (page 10, line 348) are not suitable.
- The last paragraph of the discussion section is not well supported by the indicated references (68 and 69). To the best of my knowledge, there is no published data about the immunogenicity of the boron compound that are been used for BNCT in clinical practice.
Author Response
Dear Reviewer! Thank you very much for all the critical comments. We did our best to address your doubts and critics and improve the manuscript. Below are the answers for each point.
General comment:
1. How is the selectivity of these complexes? Have they been tested in normal cells? It should be interesting to study their free uptake and accumulation in normal cells and compare it with that in tumor cells.
Unfortunately, we have not tested the activity or uptake of our nano-constructs for normal cells. In fact, the aim of the present study was to perform basic research on assembly of the 1/2 nanostructures and determination of their biological properties in cancer cells with the higher level of EGFR gene expression (as EGFR gene expression inhibition, nanostructures
cellular localization, cell cycle inhibition, inflammatory properties, free loading to tumor cells). In this study, we confirmed the certain activity of 1/2 in three types of cancer cells. Thank you for the key questions that we will try to answer in the next work. In the next step (ongoing studies) we will design oligonucleotides with nucleolytically stable modifications, and conjugate such modified tripeds with targeting molecules, which will direct ASOs to a specific types of cancer cells. Decorated with targeting molecules, the nanostructures will be
used for studies on cancerous and normal cells, which will allow the selectivity and
biodistribution of the designed oligonucleotide constructs to be assessed in more advanced models (mouse model).
2. Figure 2A: 88-bp is not trimer but tetramer
This mistake of ours has been corrected, as the 88-bp construct is made of 4 tripeds.
3. Figure 3B,C: The silencing effect of unmodified ASO-22 on the endogenous EGFR mRNA should be analysed as a reference (as it has been done in experiments shown in figures 3A an D). Thank you very much for this recommendation. We modified description of silencing activity
of nanostructures, and transferred the DFA data in A431 cells to the Supplementary Information. Instead, as recommended, we included the WB of the ASO-22 activity towards the endogenous EGFR mRNA. Thus, Fig. 3 contains additional Fig. 3B and C dealing with ASO-22, besides those Fig. 3D and Fig. 3E presenting activity of 1/2.
4. The interpretation of microscopy images of the intracellular localization of nanostructures is complex (Fig. 4). The authors state that they localize predominantly in the cytoplasm. However, fluorescent nanostructures are also clearly observed in the nucleus. By means of images shown in figure 4, it is difficult to ascertain which the main localization is. Quantification of the images could be helpful to clarify that point. We examined the localization of fluorescently labeled nanostructures in the selected cells and estimated that the nanostructures are mainly located in the cytoplasm, which is clearly
visible (i) in the FL-1 / FL-2 + DAPI row, where the nuclei are rather blue, not green. On the other hand, it is also seen that the green constructs co-localize with the ER (the bottom row). The green co-localizing fluorescence within the nucleus may indicate the presence of constructs in the cytoplasm above the nucleus and not inside the nucleus. However, we believe that demonstration of the localization of nanostructures inside the cell is very valuable, as these nanostructures exhibit inhibitory activity against the target gene. Thank you very much for pointing out the possibility of quantifying images. We will keep this
valuable note in mind in the future work. Nevertheless, in the current revision, we are not able to repeat this experiment and to obtain quantitative results due to the technical obstacles.
5. Most of the studies have been performed in A431 cells which, according to the authors, express high levels of EGFR. Why do they use Hela cells for proliferation/migration assays shown in figure 5? It should be more rational to do this study also in A431 cells. Yes, the reviewer is right that it would be better to do cancer cell phenotype and cell migration experiments in the same type of A431 cells, as these cells were used in studies of the influence of nanostructures on the cell cycle. We can only explain ourselves by the fact that the experiments shown in figure 5 were done much earlier in time than the other experiments and then we found HeLa cells to be a good model to study. HeLa cells also
express a reasonable level of EGFR protein and, as can be seen from the obtained results, lowering of the level of this protein causes the expected biological effect, i.e. a change in the morphology of HeLa cells and slowing their migration due to the reduction of EGF receptors density located at the cell surface. The results obtained on HeLa cells, in our opinion, are informative and valuable. We do not think that the activity differences of the 1/2 nanostructures between the two discussed cell lines A431 and HeLa would be significantly
different.
6. Images in Figure 5A show changes in morphology as well as in the number of cells 48 h after transfection, but they do not provide enough evidences to affirm: “More significant phenotypic changes were observed in cells transfected with 1/2 than in cells transfected with ASO-22 (Figure 5A). This result may indicate a higher intracellular activity of anti-EGFR antisense oligonucleotides present in the 1/2 nanostructures than in the case of free ASO22.” (page 8, lines 284-287). This sentence should be modified. Thank you for this suggestion. We corrected the sentence accordingly. “As shown in Figure 5A, characteristic phenotypic changes of HeLa cells including changed cellular shape and the loss of intercellular communication junctions were observed in cells transfected either with 1/2 and with ASO-22 as compared to control cells transfected with nonsense ASO-C [44,45]. This result indicates an intracellular activity of anti-EGFR antisense oligonucleotides present in the 1/2 nanostructures similar to free ASO-22..”
7. On the other hand, the wound healing assay is a cell migration assay and no conclusions about inhibition of cell proliferation must be drawn from it. There are other methods more precise and quantitative to study cell proliferation. It would also be interesting to analyse apoptosis / cell death in transfected cells.
Thank you very much the reviewer for correction of our wrong wording. Of course, the wound healing assay is informative in respect to the cell migration rate and not to the cells proliferation rate. We corrected the title and the text of section 2.6. as well as the Figure 5 caption accordingly. The changes are shown in the text with track changes version. We actually published the data on the MTT assay in HeLa cells and this result demonstrate that the toxicity of boron cluster conjugates, described as mitochondrial activity of HeLa cells, is low (Kaniowski D. et al. Nanoscale 2020).
8. Authors state that “the presence of 1,2-DCDDB in nanostructures 1/2 enhanced the antiproliferative properties of the anti-EGFR antisense oligonucleotides”, while explaining the results shown in Figure 6. As transfection with nanostructures 1/2 as well as with ASO-22 inhibit the progression to the G2 phase (2% in both cases), it would be more accurate to say that both have a similar antiproliferative ability but arrest the cell cycle in a different way (different ratio of cells in the S and in the G1 phase). How many cell cycle experiments were performed? Is there any statistical significance in these results? Thank you for this valuable comment. We improved interpretation of the results by correction the text according to the reviewer’s suggestion (section 2.7). “These results suggest that the presence of 1,2-DCDDB in nanostructures 1/2 has a small effect only on the antiproliferative properties of used anti-EGFR antisense oligonucleotides 1/2, as both tested inhibitors 1/2 and ASO-22 have a similar antiproliferative ability but arrest the cell cycle in a different way (different ratio of cells in the S and in the G1 phase).” The statistical significance of these results is presented in a new Figure presented in Supplementary Information (Fi. S10). This information is also included into the main text
(Section 2.7) and in SI (section 4.10). FACS analysis with confidence interval (CI) ) and the mean with standard deviation (±SD) statistics is given in Figure S10.
9. MTT assay is not described in the Materials and Methods section. How long were the cells incubated for the MTT assay? The following text of the MTT assay has been introduced into the section of Materials & Methods, section 4.13:
The cytotoxicity of boron clusters (1,2-DCDDB, FESAN) as well as 1/2 and ASO-22 (50 nM) in macrophages was assessed with the use of the MTT assay. Macrophages were cultured as described above. The cells were incubated for 16 h with tested compounds at 37 C under a 5% CO2 atmosphere, followed by the addition of the MTT solution in PBS (5 mg/mL) to each well. The cells were then incubated for 3 h at 37 °C under a 5% CO2 atmosphere. Finally, 95 µL of lysis buffer (NP-40, 20% SDS, 50% aqueous dimethylformamide, pH 4.5) was added to each well and cells were incubated overnight at 37 °C. The sample absorbance was measured at two wavelengths: 570 nm and the reference wavelength of 630 nm (colourless walls plate reader, PerkinElmer, Waltham, MA, USA). The results from the control cells were considered as 100% viability. The results are mean values ±SD from two technical experiments.
10. How do you explain that the boron cluster 1,2-DCDDB affects macrophages viability while there is no effect of nanostructures?
Unfortunately, we do not have a good answer to this question. It is possible that this effect occurs not only in macrophages. In general, if cell viability is decreased as a result of 1,2- DCDDB and it is inflammatory, we should expect IL-1b to be secreted as well. However, this is not the case. So, it is possible that 1,2-DCDDB does not induce pyroptosis, but rather apoptosis-resembling shutdown of metabolism, which is not pro-inflammatory. In our earlier MTT experiments (Kaniowski D et al. Nanoscale 2020 and SI to this paper) the viability of HeLa cells transfected with 1/2 (200 nM) for 48h and 72h post-transfection incubation was assessed for ca 80%, and 95 %, respectively. On the other hand we know that 1,2-DCDCB has similar effect on the cell cycle as its conjugate with oligonucleotides. Therefore we do not dare to discuss this issue in detail in the text of the paper, except to let the reviewer know that it can be that a free 1,2-DCDDB is more toxic that its conjugates with DNA.
11. Authors analyse IL-1ß production by macrophages incubated with the nanostructures to rule out an inflammatory response. Although this is a marker of inflammation, it is not the only one. I would suggest to replace the term “inflammatory/inflammation” throughout the manuscript with a more appropriate specific term. We thank the reviewer for correctly pointing out that IL-1β is not the only marker of inflammation. In this study we focused on inflammasome-dependent activation of IL-1b processing and secretion as this is a characteristic feature of terminally differentiated macrophages which we use as cellular model. We therefore corrected the term “inflammatory response/inflammation” to a more specific “inflammasome activation”.
12. Similarly, the term “immunogenicity” in line 510 (page 14) should be eliminated, as it has not been extensively studied.
We have corrected the term into more specific “pro-inflammatory properties”
13. References 51 and 52 (page 10, line 348) are not suitable.
The sentence: „Previous studies have shown that boron-containing therapeutics used for boron neutron capture therapy (BNCT) are responsible for increased levels of inflammatory cytokines in the mucosa associated with tumor tissue [51,52]” is now changed to: “Local inflammation is frequently observed during tumorigenesis. In most cancers, inflammation enhances tumor development and malignant progression [51] as well as metastasis [52], therefore potential anticancer agents should not trigger inflammatory response.”
Two references have been replaced for new ones:
Ref. 51: Todoric, J.; Antonucci, L.; Karin, M. Targeting Inflammation in Cancer Prevention and Therapy. Cancer Prev. Res. (Phila). 2016, 9(12), 895-905, doi: 10.1158/1940- 6207.CAPR-16-0209.
Ref. 52: Chow, M.T.; Sceneay, J.; Paget, C.; Wong, C.S.; Duret, H.; Tschopp, J.; Möller, A.; Smyth, M.J. NLRP3 suppresses NK cell-mediated responses to carcinogen-induced tumors and metastases. Cancer Res. 2012, 72(22), 5721-32. doi: 10.1158/0008-5472.CAN-12-0509.
14. The last paragraph of the discussion section is not well supported by the indicated references (68 and 69). To the best of my knowledge, there is no published data about the immunogenicity of the boron compound that are been used for BNCT in clinical practice. In the last paragraph we discuss the properties of boron compounds on immune and inflammatory properties. Therefore, the reference describing inflammatory response of boron
compounds in mice is cited in ref. 67, In addition, we found out some related data on boroncontaining compound in the 2018 paper (ref. 68), while the general statement on the mechanisms by which boron-containing compounds modulate immune responses, as well as discussion on the structure- activity relationship for each observed mechanism of action with respect to a production of cytokines, cell differentiation, proliferation, and antibody production were recently summarized by Romero-Aguilar K. et al. refers to new ref 69. Therefore the text has been modified accordingly (with renumeration of the references): “Boron is a regulator of the immune and inflammatory reactions and macrophage polarization, playing an important role in augmenting host defence against infection, with possible role in cancer and other diseases [68]. Boron containing compounds stimulate the production and secretion of nitric oxide, TNF-α, and IL-6, but also the secretion of IL-1β via inflammasome activation in macrophages, leading to acute inflammation in mice. Hyaboron,
a boron-containing macrodiolide, has been shown to activate inflammasome-dependent IL1β secretion by acting as potassium ionophore [69].The mechanisms by which boroncontaining compounds modulate immune responses, as well as discussion on the structureactivity relationship for each observed mechanism of action with respect to a production of cytokines, cell differentiation, proliferation, and antibody production were recently summarized by Romero-Aguilar K. et al [70].” Ref. 68: Routray, I.; Ali, S. Boron induces lymphocyte proliferation and modulates the priming effects of lipopolysaccharide on macrophages. PLOS ONE 2016, 11(3), e0150607, doi:
10.1371/journal.pone.0150607. Ref. 69: Surup, F.; Chauhan, D.; Niggemann, J.; Bartok, E.; Herrmann, J.; Keck, M.; Zander, W.; Stadler, M.; Hornung, V.; Müller, R. Activation of the NLRP3 Inflammasome by Hyaboron, a New Asymmetric Boron-Containing Macrodiolide from the Myxobacterium Hyalangium minutum. ACS Chem. Biol., 2018, 13(10), 2981-2988. doi: 10.1021/acschembio.8b00659.
Ref. 70: Romero-Aguilar, K.S.; Arciniega-Martínez, I.M.; Farfán-García, E.D.; CamposRodríguez, Rafael A.; Reséndiz-Albor A. &. Soriano-Ursúa M.A Effects of boron-containing compounds on immune responses: review and patenting trends. Expert Opinion on Therapeutic Patents, 2019, 29 (5), 339-351, DOI: 10.1080/13543776.2019.1612368

Reviewer 4 Report
This paper is devoted to the extended study of anti-EGFR antisense oligonucleotides conjugated to boron clusters (“tripeds” 1 and 2), which were further assembled into torus-like nanoparticles/ nanostructures. Synthesis and preliminary properties of these nanostructures were published previously (Kaniowski et al. Nanoscale 2020, 12, 103-114, doi: 10.1039/C9NR06550D). In the present paper authors refine probable structure of the torus-like nanoparticles assembled from 1 and 2 (1/2); characterize their nucleolytic stability; describe features of the cleavage of the [32P]-RNA target fragment of the EGFR gene hybridized with triped 1 by recombinant RNase H; confirm previously reported silencing activity of 1/2 towards the endogenous mRNA of EGFR protein in A431 cancer cells; report on the localization of fluorescently labeled 1/2 nanostructures in cancer cells; make revision of previously published preliminary data on cytotoxicity/antiproliferation activity of 1/2 nanoparticles; analyze an influence of 1/2 nanoparticles on a cell cycle of A431 cells; verify an inflammatory response of monocyte-derived macrophages to 1/2 nanoparticles; and estimate intracellular delivery of boron atoms by 1/2 nanoparticles as a basis for boron neutron capture therapy. The paper is interesting and brings important data on the anti-EGFR activity of boron- and antisense oligonucleotide- contained nanostructures. At the same time the paper requires major revision, since some experiments require additional controls and data refinement.
- From the paper it is not clear, why it is necessary to assemble 1/2 nanostructures getting a mixture of nanoparticles with different size and structure instead of using triped 1 or 2 separately. This point should be clarified.
- Conclusion about structure of “four dominant products” observed in Fig. 1A is not justified. Why, for example, was the second line assigned to tetramer, but not to trimer? Reasonable augments for these assignments are required. In the table (right side of fig. 1 A) “88bp trimer” is written, but the structure of tetramer is shown.
- Lines 214-217. It is better to write that the silencing activity of 1/2 was found and reported in [ref35] and it is confirmed in the present paper by fluorescence microscopy. Since it is not a new result, Fig. 3D should be presented in Supplementary information.
- Lines 261-262. “This process involves, at least in part, helicase-assisted unwinding of nanostructures 1/2 to their individual strands”. This statement should be proved or reduced to “probably”.
- Line 265. “some amount of nanostructured fluorescent B-ASO is transported into the nuclei” – this conclusion should be carefully verified. DAPI fluorescence can mix with fluorescence of fluorescein in FL-1/FL-2 and be a reason of nucleus-associated signal. It is necessary to describe the used filter sets in Materials and methods. And it is necessary to make control experiments: (1) stain cells with DAPI only and measure fluorescent image of cells with a filter set used for fluorescein detection to verify contribution of DAPI signal in fluorescein images; (2) stain cells with FL-1 (or FL-2) only and check for the presence of FL-1 (or FL-2) in nucleus (transmitted light images can be used to recognize positions of nuclei in cells).
- subsection 2.6. Data presented in this subsection are in contradiction with the data obtained previously [35] on Hela cells with MTT assay. MTT demonstrated no effect of complexes 1/2 on HeLa cells, whereas new data support antiproliferation action and, possibly, cytotoxicity of 1/2. Antiproliferation effect should be easy detected with MTT, since the number of cells becomes smaller as compared to control. This contradiction should be explained.
It is not clear, why effects of 1/2 complexes on proliferation and cell cycle were studied for HeLa cells, while other activities were reasonably investigated for A431 cells with super-expression of EGFR.
- Figure 5B, lines 295-306. A wound healing assay is a technique to study cell migration, and when a considerable antiproliferation effect is observed, this technique cannot discriminate influence of tested compounds on cell migration (except for strong enhancement of cell migration). Accordingly, Figure 5B just confirms results shown in Fig. 5A. I propose to remove Fig.5B and corresponding text. Instead MTT or XTT test could help to quantify reliably antiproliferative effect of tested compounds. In addition, staining of treated cells with DAPI and propidium iodide followed by fluorescence microscopy examination could reveal probable cytotoxicity aroused, for example, after the reported cell cycle arrest.
- Subsection 2.7 and Fig. 7. (a) Control measurement describing effect of Lipofectamine on a cell cycle is required.
(b) The presented results prove effect of ASO-22, 1/2 complexes and 1,2-DCDDB on cell cycle, but do not clarify an origin of this effect in the case of 1/2 complexes. Boron cluster or antisense oligonucleotide is responsible? Effect is similar at 50 and 100 nM. Probably a saturation of effect was achieved, and lower concentrations of compounds are required to reveal the difference in effect for boron clusters and oligonucleotides.
- Subsection 2.9. Very efficient intracellular penetration of 1/2 complexes without Lipofectamine, which is followed from the reported high boron content in cells, is very intriguing. This makes meaningless all the reported results obtained with Lipofectamine.. Accordingly, this result requires independent confirmation. For example, cells can be incubated with fluorescently labeled 1/2 complexes to see their assumed internalization without transfection.
Minor remarks.
All abbreviations should be introduced in the text, when they appear for the first time.
For example, ASOs (line 67), LCA CPG (line 109), RT (line 562).
Line 80. It is written “in this paper”, but the reference is indicated to the previous paper.
Line 99. “the assembly of tripeds 1 and 2” – a reference to the Figure 1 and a brief description of these assemblies are required.
Line 172 abbreviation “B-ASO” is not required.
Paragraph 2.4.3. – Different abbreviations are used for the same construction: EGFR-GFP (line 215), EGFR-EGFP (line 220), pEGFR-EGFP (lines 229, 240). Please, select one.
Lines 247-248. “The procedures for the synthesis…. provided in the Supplementary Information (Figure S1).” No description of synthesis in the Supplementary information.
Line 256. “green fluoroscopy imaging” change to fluorescence microscopy
Line 273. “All panels were enlarged 60-times” – Scale bars should be shown in images.
Lines 281-283. “Characteristic phenotypic changes of HeLa cells were… compared to control cells”. It is just a general statement, and no observed phenotypic changes are described.
Line 328. FESAN- what is this, and why was it used in the studies?
Lines 342-344. Figure legend does not describe all data presented in figure.
Lines 364-371 MTT is a cell viability assay according to panel B, but it is described as a mitochondrial activity assay in Figure 7 legend. Use “cell viability assay” everywhere.
Subsection 4.13. (a) Conditions of the experiment without Lipofectamine are not described.
(b) Why was it necessary to continue incubation of cells for 48 h after removal of the transfection mixture? It seems that immediately after removal of the transfection mixture the intracellular concentration of boron was the highest, and during next 48 h it probably decreased because of nanoparticle efflux.
Lines 452-453. This sentence, seems, has no relation to the paper subject.
Author Response
The paper is interesting and brings important data on the anti-EGFR activity of boron- and antisense oligonucleotide- contained nanostructures. At the same time the paper requires major revision, since some experiments require additional controls and data refinement
Dear Reviewer!
Thank you very much for all the critical comments. We did our best to
address your doubts and critics and improve the manuscript. Below are the answers for each point.
- From the paper it is not clear, why it is necessary to assemble 1/2 nanostructures getting a mixture of nanoparticles with different size and structure instead of using triped 1 or 2 separately. This point should be clarified.
We are sorry that the description in the publication was not clear enough, although we wrote that triped1 is composed of two oligos antisense to mRNA of EGFR. In contrast triped 2 is complementary to this triped 1. So, the separate delivery of 1 and of 2 would not offer any added value to the common antisense oligonucleotide (as ASO-22 here). Only a 1:1 complex of both tripeds, which we proved to adopt ring-shape nanostructures, is attractive as a new scaffold for antisense approach. These nanostructures, visualized by cryo-TEM and AFM
imaging in our preliminary data published in 2020 in Nanoscale, are ca 20 nm in size which offers easy uptake by the cells (Fig 8, free uptake of 1/2 as shown in this work by ICP-MS). Moreover, we demonstrate here that nanostructures with embodied clusters and oligo strands are nucleolytically more stable than ASO-22 (Fig. 2), and also more active in EGFR downregulation (WB assay). So, in our opinion this assembly of 1/2 is a crucial step in our approach. - Conclusion about structure of “four dominant products” observed in Fig. 1A is not justified. Why, for example, was the second line assigned to tetramer, but not to trimer? Reasonable augments for these assignments are required. In the table (right side of fig. 1 A) “88bp trimer” is written, but the structure of tetramer is shown. Thank you for pointing out our mistake. 88-bp complex consists of four tripeds (two tripeds 1 and two tripeds 2), as it is schematically shown in the Table of Fig. 2A. This experiment has shown that four complexes are formed in solution, and their “size” is assigned as 44-bp (two tripeds, 1x1 + 1x2), 88-bp (four tripeds, 2x1 + 2x2), 132-bp (six tripeds, 3x1 + 3x2) and 176
(eight tripeds, 4x1 + 4x2). Since we have already demonstrated that such complexes form ring-type structures (Cryo-TEM, AFM), due to their complementarity and the determined structure of the boron cluster (substituents in boron cluster are located at the angle of ca 60 degrees), the number of tripeds has to be even.
3. Lines 214-217. It is better to write that the silencing activity of 1/2 was found and reported in [ref35] and it is confirmed in the present paper by fluorescence microscopy. Since it is not a new result, Fig. 3D should be presented in Supplementary information. Yes, indeed, this result is a repetition of our data from ref. 35. Thank you very much for suggestion to present former Fig. 3D in Supplementary information. This was done as recommended, and the text in the main body was corrected accordingly. In addition, we demonstrate here (Fig. 3C and D) the silencing activity of ASO-22 measured towards endogenous EGFR by Western blot imaging.
4. Lines 261-262. “This process involves, at least in part, helicase-assisted unwinding of nanostructures 1/2 to their individual strands”. This statement should be proved or reduced to “probably”. This expression has been corrected as suggested.
5. Line 265. “some amount of nanostructured fluorescent B-ASO is transported into the nuclei” – this conclusion should be carefully verified. DAPI fluorescence can mix with fluorescence of fluorescein in FL-1/FL-2 and be a reason of nucleus-associated signal. It is necessary to describe the used filter sets in Materials and methods. And it is necessary to make control experiments: (1) stain cells with DAPI only and measure fluorescent image of cells with a filter set used for fluorescein detection to verify contribution of DAPI signal in
fluorescein images; (2) stain cells with FL-1 (or FL-2) only and check for the presence of FL1 (or FL-2) in nucleus (transmitted light images can be used to recognize positions of nuclei in cells). We performed two control experiments as suggested, and the respective data are shown in Supplementary information (Figure S8). We conclude that there is no mixing of signals of FITC and DAPI.
Figure S9. DAPI and FITC filters test demonstrating no emission signals of FITC and DAPI due to the DAPI and FITC fluorophores excitation, respectively. (A) A431 cells transfected with FL-1/FL2 (4 µM) cells in the presence of lipofectamine 2000 and analyzed with the FITC filter (exposure time 1s) and checked with the DAPI filter (1s and 2s); (B) A431 cells treated with DAPI (5 µg/mL) and analyzed using a DAPI filter (exposure time 300ms) and checked
using a FITC filter (exposure time 300ms and 1s). As a consequence, we also reinterpreted our data and corrected the text in the main text, as follow:
However, slightly fewer blue nuclei in row 4 and the presence of green nuclei in row 5 might originate from the overlap of green and blue fluorescence emission signals (see Fig. S9) rather due to presence of the fluorescent B-ASO nanostructures over the nuclei than from the presence of FL-1/FL-2 inside of nuclei, as it was shown by previous studies for antisense
oligonucleotides [47]. The used filter sets in are described Materials & Methods, section 4.11 as follow: After incubation, the cells were washed two times with PBS buffer (with Ca2+ and Mg2+) and imaged under a fluorescence microscope (Nikon-Eclipse, Japan) to detect DAPI at λex = 340-380 nm (λDM = 400 nm, λBA = 435-485 nm) and FITC at λex = 465-495 nm (λDM = 505 nm, λBA = 515-555 nm), B-2A (longpass, λex = 450-490 nm(λDM = 505 nm, λBA = 520 nm), TX red (λex = 540-580 nm, λDM =595 nm, λBA = 600-660 nm), and G-2A longpass λex = 510-560 nm(λDM = 575 nm, λBA = 590 nm), where DM is a dichroic mirror and BA is an absorption filter.
6. subsection 2.6. Data presented in this subsection are in contradiction with the data obtained previously [35] on Hela cells with MTT assay. MTT demonstrated no effect of complexes 1/2 on HeLa cells, whereas new data support antiproliferation action and, possibly, cytotoxicity of 1/2. Antiproliferation effect should be easy detected with MTT, since the number of cells becomes smaller as compared to control. This contradiction should be explained. Thank you for this valuable and key criticism. The test we performed was named incorrectly
in some parts of the work, for which we apologize. The test described here is a test for cell migration or cell mobility, and not a proliferation test. The wound healing assay, used to assess cells migration, showed that as a result of a reduction level of endogenous EGFR by 1/2 in the cancer cells, cell migration in the wound area was inhibited, compared to the control cells. If the number of cells changed as a result of the proliferation process, we would observe these changes in the cytotoxicity assay (MTT), what is not a case (ref. 35). Also in
the present work in the 2.8 section (Fig. 7B) we demonstrate that nanostructures 1/2 are not cytotoxic to tested cells (macrophages) in the MTT assay. The text of the section 2.6 has been corrected accordingly and the migration rate wording was properly used instead of “proliferation”.
7. It is not clear, why effects of 1/2 complexes on proliferation and cell cycle were studied for HeLa cells, while other activities were reasonably investigated for A431 cells with superexpression of EGFR. The cell cycle experiment was performed on A431 cells and not on HeLa cells. However, the cell migration assay was done on HeLa cells and it was done much earlier in the course of the studies, when we tested properties of 1/2 on the HeLa cells. The results shown in earlier work (ref. 35) obtained in the DFA model showed that the nanostructures in both the HeLa and A431 lines are equally active against exogenous EGFR after 48 hours, what indicates
similar properties of 1/2 in different types of cancer cells. On the other hand we do not think that the 1/2 activity between the two lines A431 and HeLa is extremely different. Therefore, we suggest that the presented tests are sufficiently informative on the properties of the studied complexes of 1/2.
8. Figure 5B, lines 295-306. A wound healing assay is a technique to study cell migration, and when a considerable antiproliferation effect is observed, this technique cannot discriminate influence of tested compounds on cell migration (except for strong enhancement of cell migration). Accordingly, Figure 5B just confirms results shown in Fig. 5A. I propose to remove Fig.5B and corresponding text. Instead MTT or XTT test could help to quantify reliably antiproliferative effect of tested compounds. In addition, staining of treated cells with
DAPI and propidium iodide followed by fluorescence microscopy examination could reveal probable cytotoxicity aroused, for example, after the reported cell cycle arrest. In answer for point #5 we elaborated in detail over the issue dealing with the wound healing assay experiment, where we examined cell migration rather than cells proliferation. Therefore, the conclusions on the antiproliferative effect of our nanostructures are rather inadequate. At the same time, we consider both results presented in Figure 5 to be valuable as the test for assessing the phenotypic changes in cells after EGFR gene silencing (Fig. 5A)
is different from the migration test (Fig. 5B). We propose to leave Fig. 5 as is, while the data on MTT are included in [35] and in this paperon macrophages (Fig. 7B).
9. Subsection 2.7 and Fig. 7. (a) Control measurement describing effect of Lipofectamine on a cell cycle is required. First, we would like to clarify that for the control experiment of cell cycle analysis we used A431 cells treated with lipofectamine in the amount corresponding to transfection of A431 cells with 100 nM 1/2. Therefore, this corrected data was introduced into the text in
Section 2.7 and in Materials & Methods section 4.10. Moreover, we repeated the control experiment in which we compared cell cycle of A431 cells without and with lipofectamine. The results indicate only minor differences between both analyses (higher by 3% in G0/G1 phase and lower by 3% in G2/M phase, and no difference in the S phase upon lipofectamine treatment).
(b) The presented results prove effect of ASO-22, 1/2 complexes and 1,2-DCDDB on cell cycle, but do not clarify an origin of this effect in the case of 1/2 complexes. Boron cluster or antisense oligonucleotide is responsible? Effect is similar at 50 and 100 nM. Probably a saturation of effect was achieved, and lower concentrations of compounds are required to reveal the difference in effect for boron clusters and oligonucleotides. The saturation effect is an interesting explanation for the similar cell cycle results for 1/2 used
at 50 and 100 nM. Unfortunately, we did not perform any cell cycle experiment at the lower concentration because the 1/2 silencing effect applied at the lower concentration (25 nM) was low (only a 20% decreased level of endogenous EGFR was observed, as shown in Fig. 3B). In our opinion, the observed effect of the cell cycle phases after transfecting A431 cells with 1/2 is due to both the antisense activity of the oligonucleotide (19% in S phase) and the 1,2-DCDDB (22% in S phase), as a total of 30% of cells were in S phase when 1/2 was used at 50 nM compared, to 16% for A431 control cells.
10. Subsection 2.9. Very efficient intracellular penetration of 1/2 complexes without Lipofectamine, which is followed from the reported high boron content in cells, is very intriguing. This makes meaningless all the reported results obtained with Lipofectamine. Accordingly, this result requires independent confirmation. For example, cells can be incubated with fluorescently labeled 1/2 complexes to see their assumed internalization without transfection.
Thank you for appreciating our results presented in point 2.9. Although the 1/2 cellular uptake is still twice as high with the help of Lipofectamine, we have proven that the delivery of nanostructures without transfection agent is somewhat effective (50% transfection level with Lipofectamine). This result is shown here by ICP MS (Figure 8). The reviewer asked for independent confirmation of this result, suggesting checking for the internalization of
fluorescently labeled 1/2. We have actually already performed such experiments and confirmed this internalization with FL-1 / FL-2 on the similar level as obtained by ICP MS. The mechanism of this free cellular uptake is currently being investigated and we need more data to publish it in an independent article.
11. All abbreviations should be introduced in the text, when they appear for the first time. For example, ASOs (line 67), LCA CPG (line 109), RT (line 562).
All the abbreviations were defined along the text.
12. Line 80. It is written “in this paper”, but the reference is indicated to the previous paper. Corrected accordingly. In our previous research, we designed novel self-assembling DNA nanostructures bearing antisense oligonucleotides directed towards EGFR mRNA [35].
13. Line 99. “the assembly of tripeds 1 and 2” – a reference to the Figure 1 and a brief description of these assemblies are required. The assembly of tripeds 1 and 2 to the dimer 1/2 is shown at corrected Fig. 1, and the caption clarifies the binding scheme.
14. Line 172 abbreviation “B-ASO” is not required. Corrected accordingly.
15. Paragraph 2.4.3. – Different abbreviations are used for the same construction: EGFRGFP (line 215), EGFR-EGFP (line 220), pEGFR-EGFP (lines 229, 240). Please, select one. Corrected to EGFR-EGFP/RFP model.
16. Lines 247-248. “The procedures for the synthesis…. provided in the Supplementary Information (Figure S1).” No description of synthesis in the Supplementary information. The procedure description is provided in the main text and the above sentence was corrected accordingly. “The procedure for the synthesis of FL-1 and FL-2 is described in section 4.1 and the chromatographic and structural analyses of FL-1 and FL-2 are provided in the Supplementary Information (Figures S1-S3).”
17. Line 256. “green fluoroscopy imaging” change to fluorescence microscopy
Thank you for such careful correction! Corrected accordingly.
18. Line 273. “All panels were enlarged 60-times” – Scale bars should be shown in images. Here, the scale bar is the same for each picture in a given column, as all pictures in a given column refer to the same field viewed.
19. Lines 281-283. “Characteristic phenotypic changes of HeLa cells were… compared to control cells”. It is just a general statement, and no observed phenotypic changes are described. The text was corrected accordingly, as follow: As shown in Figure 5A, characteristic phenotypic changes of HeLa cells, as changed cellular shape, and the loss of intercellular communication junctions were observed in cells transfected either with 1/2 and with ASO-22 as compared to control cells transfected with nonsense ASO-C [44,45].
20. Line 328. FESAN- what is this, and why was it used in the studies?
It is explained in the text (section 2.7) that: “a control boron cluster, a metallacarborane [(3,3′- iron-1,2,1′,2′-dicarbollide) (-1)]-ate [Fe(C2B9H11)2]- (FESAN))”.
21. Lines 342-344. Figure legend does not describe all data presented in figure.
Figure legend has been improved as suggested. Figure 6. FACS analysis of the cell cycle phases of A431 cells upon Lipofectamine 2000- assisted transfection with ASO-22 and nanostructures 1/2 (50 and 100 nM) after 48 h of
incubation, and for A431 cells treated with 1,2-DCDDB and FESAN (50 and 100 nM). The A431 cells treated with Lipofectamine 2000 only were used as a control. Colchicine and quercetin were used as positive controls. ****P ≤ 0.0001 (ANOVA).
22. Lines 364-371 MTT is a cell viability assay according to panel B, but it is described as a mitochondrial activity assay in Figure 7 legend. Use “cell viability assay” everywhere. The text has been corrected accordingly, and the term “mitochondrial activity assay” was replaced either with “cytotoxicity” or with “cell viability assay”.
23. Subsection 4.13. (a) Conditions of the experiment without Lipofectamine are not described. The text of section (now) 4.14 has been completed as recommended: “In order to conduct the experiment for the assessment of the free uptake of 1/2, the medium was replaced to basic medium (DMEM containing 4.5 g/L D-glucose, 0.11 g/L sodium pyruvate and without L-glutamine (Gibco, BRL, Paisley, New York, NY, USA), 100 U/mL penicillin, and 100 µg/mL streptomycin (Gibco, BRL, Paisley, New York, NY, USA) at 37 C
and 5% CO2) and the aqueous solutions of 1/2 (0.1, 0.2, 0.4, 0.8, 2, and 4 µM) was added to the medium and incubated for 48h at 37 C in an atmosphere of 5% CO2. After 48 h (experiments with or without Lipofectamine 2000) and medium removal….” (b) Why was it necessary to continue incubation of cells for 48 h after removal of the transfection mixture? It seems that immediately after removal of the transfection mixture the intracellular concentration of boron was the highest, and during next 48 h it probably decreased because of nanoparticle efflux. We have shown that 1/2 nanostructures exert silencing effect after 48 hours of incubation after transfection. Therefore, we measured the boron content in the same time. It is already published that radiotherapy and BNCT are more effective in cells with decreased level of EGFR - Ref [59, 60]. Therefore it was rather rational to orchestrate these two experiments. However, due to their high lipophilicity, boron clusters can bind to cell membranes, proteins
and organelles, which makes their potential escape difficult. Moreover, they are negatively charged, and this feature also prevents their efflux. In contrast, boron-containing compounds conjugated to a hydrophilic molecule may exhibit a high efflux rate.
24. Lines 452-453. This sentence, seems, has no relation to the paper subject.
The sentence: "The strong affinity of other boron clusters, including metallacarborane (FESAN), to snake venom proteins, was also demonstrated by microscale thermophoresis [63].” has been deleted.
The numbers of references were adjusted.

Round 2
Reviewer 3 Report
Authors have addressed most comments though few minor points could be still improved:
- The explanation about nuclear fluorescence is confusing (page 7, lines 271-275). It should be rewrite for further clarification.
- The new sentence in section 2.7. (page 10, lines 336-337) must also be revised.
- Figure 6 does not include any statistical analysis. Hence, the last sentence in figure caption “****P ≤ 0.0001 (ANOVA and post-hoc Tukey HSD test)” must be deleted.
- Reference 66 in line 514 (page 14) must be replaced with reference 67.
Author Response
- The explanation about nuclear fluorescence is confusing (page 7, lines 271-275). It should be rewrite for further clarification.
However, slightly fewer blue nuclei in row 4 and the presence of green nuclei in row 5 rather due to presence of the fluorescent B-ASO nanostructures over the nuclei than from the presence of FL-1/FL-2 inside of nuclei, as it was shown by previous studies for antisense oligonucleotides [47].
Now the text has been reworded to:
Moreover, the green signals overlapping the DAPI signal in the 4th row, and the green signals next to the ER signals in the 5th row suggest that FL-1 / FL-2 may also be present in the nuclei. There are studies on the PS-ASOs which showed the nuclear localization of antisense oligonucleotides [47]. As shown in the DAPI and FITC filter test (see Fig. S9) the observed green signals in the nuclei are not from excitation of the fluorescein chromophore by the blue wavelength emitted by DAPI.
- The new sentence in section 2.7. (page 10, lines 336-337) must also be revised.
These results suggest that the presence of 1,2-DCDDB in nanostructures 1/2 has a small effect only on the antiproliferative properties of used anti-EGFR antisense oligonucleotides 1/2, as both tested inhibitors 1/2 and ASO-22 have a similar antiproliferative ability but arrest the cell cycle in a different way (different ratio of cells in the S and in the G1 phase).
Now the text has been reworded to:
The changes observed in a cell cycle phases ratio induced by ASO-22 and 1/2 follow a very similar pattern to 1,2-DCDDB, with small changes only in the S (increased by 3-14%) and G1 phases (-1÷10%) compared to the control cells, while the abundance of the G2 phase remains very low in all experiments (2-3%) compared to 15% for the control cells.
- Figure 6 does not include any statistical analysis. Hence, the last sentence in figure caption “****P ≤ 0.0001 (ANOVA and post-hoc Tukey HSD test)” must be deleted.
Corrected accordingly.
- Reference 66 in line 514 (page 14) must be replaced with reference 67.
Corrected accordingly.

Reviewer 4 Report
Authors improved the manuscript and responded to my notes.
I have only minor notes to the new version of the manuscript:
Lines 217-276. The sentence is too long and complicated. I can not undersand the following part of the sentence: "... originate from the overlap of green and blue fluorescence emission signals (see Fig. S9) rather due to presence of the fluorescent B-ASO nanostructures over the nuclei than from the presence of FL-1/FL-2 inside of nuclei... "
Line 460 "nanostructures 1/2 () compared" - delete ().
Line 461 "(t½ = t½ = 45 and 21 h, respectively)" - delete repeated "t½ =".
Line 645 correct "@c".
Line 745 "is an absorption filter" - change to "is an emission filter"
Author Response
Lines 217-276. The sentence is too long and complicated. I can not undersand the following part of the sentence: "... originate from the overlap of green and blue fluorescence emission signals (see Fig. S9) rather due to presence of the fluorescent B-ASO nanostructures over the nuclei than from the presence of FL-1/FL-2 inside of nuclei... " The text has been corrected as follow: Moreover, the green signals overlapping the DAPI signal in the 4th row, and the green signals next to the ER signals in the 5th row suggest that FL-1 / FL-2 may also be present in the nuclei. There are studies on the PS-ASOs which showed the nuclear localization of antisense oligonucleotides [47]. As shown in the DAPI and FITC filter test (see Fig. S9) the observed green signals in the nuclei are not from excitation of the fluorescein chromophore by the blue wavelength emitted by DAPI. Line 460 "nanostructures 1/2 () compared" - delete (). Corrected accordingly. Line 461 "(t½ = t½ = 45 and 21 h, respectively)" - delete repeated "t½ =". Corrected accordingly. Line 645 correct "@c". Corrected accordingly. Line 745 "is an absorption filter" - change to "is an emission filter" Corrected accordingly.
